# Adaptive Exploration for Data-Efficient General Value Function Evaluations

**Arushi Jain**
arushi.jain@mail.mcgill.ca
McGill University
Mila

**Josiah P. Hanna**
jphanna@cs.wisc.edu
The University of Wisconsin – Madison

**Doina Precup**
dprecup@cs.mcgill.ca
McGill University
Mila

## Abstract

General Value Functions (GVFs) (Sutton et al., 2011) represent predictive knowledge in reinforcement learning. Each GVF computes the expected return for a given policy, based on a unique reward. Existing methods relying on fixed behavior policies or pre-collected data often face data efficiency issues when learning multiple GVFs in parallel using off-policy methods. To address this, we introduce `GVFExplorer`, which adaptively learns a single behavior policy that efficiently collects data for evaluating multiple GVFs in parallel. Our method optimizes the behavior policy by minimizing the total variance in return across GVFs, thereby reducing the required environmental interactions. We use an existing temporal-difference-style variance estimator to approximate the return variance. We prove that each behavior policy update decreases the overall mean squared error in GVF predictions. We empirically show our method's performance in tabular and nonlinear function approximation settings, including Mujoco environments, with stationary and non-stationary reward signals, optimizing data usage and reducing prediction errors across multiple GVFs.

## 1  Introduction

The ability to make multiple predictions is a key attribute of human, animal, and artificial intelligence. Sutton et al. (2011) introduced **General Value Functions (GVFs)** which consists of several independent sub-agents, each responsible for answering specific predictive knowledge about the environment. Each GVF consists of a unique - policy, custom reward function called *cumulant* and state-dependent discount factor - to calculate the cumulative discounted cumulant. For example, a GVF can predict the expected number of times an agent will bump into the wall under a given policy (White et al., 2015; Schlegel et al., 2021; Sherstan, 2020). In essence, GVFs generalizes the standard value function to address a wider range of predictive questions, making them a powerful tool for intelligent systems.

Prior works have used either a fixed random behavior policy (Sutton et al., 2011) or pre-collected datasets (Xu et al., 2022) to update all GVFs in parallel using *off-policy learning*. However, these methods can result in large value estimation errors if the behavior policy significantly diverges from the GVF policies. Our work addresses this gap by focusing on the question of exploration for evaluating GVFs: *how can we adapt an agent's behavior policy for data-efficient estimation of multiple GVFs in parallel?* While exploration has been extensively studied in the context of optimal control in Markov Decision Processes (MDP), the question of constructing a policy that can learn multiple value functions in parallel has remained largely unexplored.

To accurately evaluate multiple GVFs in parallel, we aim to design a behavior policy that minimizes the overall *mean squared error (MSE)* of their predictions. A natural approach might involve following each GVF's target policy for some period of time (e.g. one episode) in a round-robin manner while concurrently updating all GVFs off-policy. However, this approach can be highly

data inefficient as actions are sampled according to given policies, potentially overlooking actions from states where the expected return is uncertain. To achieve better value estimation with fewer samples, it is essential to focus on state-action pairs with high variance in return, as these pairs would exhibit greater uncertainty in their mean return. Therefore, a behavior policy should visit such pairs more frequently to offset higher variance in return. Consider an analogy of a two-arm bandit problem: fewer samples are needed to accurately evaluate a constant reward arm, whereas an arm with a variable reward demands a greater number of samples to achieve the same level of certainty. We empirically support this claim by comparing round-robin and our approach later in this paper.

With this motivation, we introduce `GVFExplorer`, that adaptively learns a behavior policy which minimize the total MSE across all GVF predictions. `GVFExplorer` leverages the existing off-policy temporal difference (TD) based estimator of variance in return distribution (Sherstan et al., 2018; Jain et al., 2021) to guide the behavior policy. The strategy is to frequently take actions that might have more unpredictable outcomes (high variance in return). By sampling them more, agent can estimate the mean return better with fewer interactions, thus effectively lowering the overall MSE in the GVF predictions.

`GVFExplorer` optimizes the data usage and reduces the prediction error, offering a scalable solution for complex environments. This is particularly valuable for real-world applications like personalized recommender systems (Parapar & Radlinski, 2021; Tang et al., 2015), where it can enable efficient evaluation of personalized policies based on diverse user preferences (reward functions) (Li et al., 2024), leveraging shared knowledge for improved accuracy.

**Contributions:** (1) We design an adaptive behavior policy that enables accurate and efficient learning of multiple GVF predictions in parallel [Algorithm 1]. (2) We derive an iterative behavior update rule that directly minimizes the overall prediction error [Theorem 4.1]. (3) We prove in the tabular setting that each iterative update to the behavior policy causes the total MSE across GVFs to be less than equal to one from the old policy [Theorem 4.2]. (4) We establish the existence of a variance operator that enables us to use TD-based variance estimation [Lemma 5.1]. (5) We empirically demonstrate in both tabular and Mujoco environments that `GVFExplorer` lowers the total MSE when estimating multiple GVFs compared to baseline approaches and enables evaluating a larger number of GVFs in parallel.

## 2 Related Work

Exploration in reinforcement learning (RL) has predominantly focused on improving policy performance for a single objective (Oudeyer et al., 2007; Schmidhuber, 2010; Jaderberg et al., 2016; Machado et al., 2017; Eysenbach et al., 2018; Burda et al., 2018; Guo et al., 2022). Refer to Ladosz et al. (2022) for a detailed survey on exploration techniques in RL. While related to exploration, these works differ from ours, as they concentrate on optimizing policies for single objective rather than evaluating multiple GVFs (policy-cumulant pair) simultaneously.

Our work is most closely related to other works on learning multiple GVFs. Xu et al. (2022) address a similar problem by evaluating multiple GVFs using an offline dataset, but our method operates online, avoiding the data coverage limitations of offline approaches. Linke et al. (2020) develops exploration strategies for GVFs in a stateless bandit context, which does not deal with the off-policy learning or function approximation challenges present in the full Markov Decision Process (MDP) context. In a single bandit problem, Antos et al. (2008); Carpentier et al. (2015), show that the optimal data collection strategy to estimate mean rewards of arms is to sample proportional to each arm's variance in reward. Prior works like Hanna et al. (2017) learned a behavior policy for a *single* policy evaluation problem using a REINFORCE-style (Williams, 1992) variance-based method called BPS. This idea extends on the similar principles of using Importance Sampling in Monte Carlo simulations for finding optimal sampling policy based on variance minimization (Owen, 2013; Frank et al., 2008). Metelli et al. (2023) extends this idea to the control setting. However, these methods are limited to single-task evaluation or control. Evaluating multiple policies simultaneously is more complex, requiring careful balance in action selection among interrelated learning problems. Perhaps the closest work to ours is by McLeod et al. (2021), which uses the changes in the weights of Successor Representation (SR) (Dayan, 1993) as an intrinsic reward to learn a behavior policy that supports multiple predictive tasks. `GVFExplorer` approach is simpler, as it directly optimizes the behavior

policy to minimize the total prediction error over GVFs, resulting in an intuitive variance-proportional sampling algorithm. We will compare the two approaches empirically as well.

## 3 Preliminaries

Consider an agent interacting with the environment to obtain estimates of $N$ different *General Value Function* (GVF) (Sutton et al., 2011). We assume an episodic, discounted Markov decision process (MDP) where $\mathcal{S}$ is the set of states, $\mathcal{A}$ is the action set, $\mathcal{P} : \mathcal{S} \times \mathcal{A} \to \Delta_{\mathcal{S}}$ is the transition probability function, $\Delta_{\mathcal{S}}$ is the $|\mathcal{S}|$-dimensional probability simplex, and $\gamma \in [0, 1)$ is the discount factor.

Each GVF is conditioned on a fixed policy $\pi_i : \mathcal{S} \to \Delta_{\mathcal{A}}$, $i = \{1, \dots, N\}$ and has a cumulant $c_i : \mathcal{S} \times \mathcal{A} \to \mathbb{R}$. For simplicity, we assume that all cumulants are scalar, and that the GVFs share the environment discount factor $\gamma$. This eases the exposition, but our results can be extended to general multidimensional cumulants and state dependent discount factor. Each GVF is a value function $V_{\pi_i}(s) = \mathbb{E}_{\pi_i, \mathcal{P}}[G_t^i | s_t = s]$ where $G_t^i = c_{i,t} + \gamma G_{t+1}^i$. Each GVF can be viewed as answering the question, "what is the expected discounted sum of $c_i$ received while following $\pi_i$?" We can also define action-value GVFs: $Q_{\pi_i}(s, a) = c_i(s, a) + \gamma \mathbb{E}_{s' \sim \mathcal{P}(\cdot|s,a)}[V_{\pi_i}(s')]$, with $V_{\pi_i}(s) = \mathbb{E}_{a \sim \pi_i(\cdot|s)}[Q_{\pi_i}(s, a)]$.

At each time step $t$, the agent in state $s_t$, takes an action $a_t$ and receives cumulant values $c_{i,t}$ for all $i \in \{1, \dots, N\}$, transitioning to a new state $s_{t+1}$. This repeats until reaching a terminal state or a maximum step count. Then the agent resets to a new initial state and starts again. The agent interacts with environment using a behavior policy, $\mu : \mathcal{S} \to \Delta_{\mathcal{A}}$. The goal is to approximate values $\hat{V}_i$ corresponding to the true GVFs value $V_{\pi_i}$. We formalize the objective as **minimizing the Mean Squared Error (MSE)** under some state weighting $d(s)$ for all GVFs:

$$MSE(V, \hat{V}) = \sum_{i=1}^{N} \sum_{s \in \mathcal{S}} d(s) \Big( V_{\pi_i}(s) - \hat{V}_i(s) \Big)^2. \tag{1}$$

In our experiments, we use the uniform distribution for $d(s)$. This objective can be generalized to prioritize certain GVFs using a weighted MSE.

**Importance Sampling (IS).** To estimate multiple GVFs with distinct target policies $\pi_i$ in parallel, off-policy learning is essential. Importance sampling (IS) is one of the primary tools for off-policy value learning (Hesterberg, 1988; Precup, 2000; Rubinstein & Kroese, 2016), allowing estimation of value function under target policy $\pi$ using samples from different behavior policy $\mu$. In the context of off-policy Temporal Difference (TD) learning (Sutton & Barto, 2018), the IS ratio, $\rho_t = \frac{\pi(a_t|s_t)}{\mu(a_t|s_t)}$, is used to adjust the updates to ensure *unbiased value estimates*. The update rule is given as $\hat{Q}(s_t, a_t) = \hat{Q}(s_t, a_t) + \alpha \left( c_t + \gamma \rho_{t+1} \hat{Q}(s_{t+1}, a_{t+1}) - \hat{Q}(s_t, a_t) \right)$, where $\alpha$ is the learning rate.

This update rule ensures that estimated value function $\hat{Q}$ converges to correct value $Q_\pi$ under policy $\pi$, despite the samples being generated from a behavior policy $\mu$.

## 4 Behavior Policy Optimization

As described in the previous section, the goal of the agent is to minimize the total mean squared error (MSE) across the given GVFs (Eq. (1)). Note that MSE = Variance + Bias$^2$. For the algorithm's derivation, we will use **unbiased** IS estimation for off-policy correction, which shifts the task of minimizing MSE to reducing the total variance across GVFs. *Thus, the core problem is to design a behavior policy that collects data to **minimize the variance in return** across all GVFs in order to accurately estimate multiple GVF value functions.*

The problem of estimating a *single target policy's* value, $V_\pi$, is well studied in the literature (see Sec. 2). In Monte Carlo sampling literature, it is well known that there exists an optimal sampling distribution (i.e., behavior policy) that provides optimal variance reduction compared to simply running the target policy Kahn & Marshall (1953); Owen (2013). Unfortunately, the analytical solution of obtaining this optimal behavior policy, $\mu^*$, requires foreknowledge of $V_\pi$, making it impractical when our overall purpose is to estimate $V_\pi$. Nonetheless, in this work, we take inspiration from this earlier work and develop a practical method that iteratively solves for a single behavior policy that minimizes the total variance when estimating multiple general value functions in parallel.

## 4.1 Objective Function

We propose `GVFExplorer` to address the above limitation and extend the problem to accurately estimate multiple GVF values. `GVFExplorer` takes as input the GVF target policies $\pi_{i=\{1,\ldots,N\}}$, collects data from a single (non-stationary) behavior policy $\mu$, and outputs the GVF estimates $\hat{V}_{i=\{1,\ldots,N\}}$. Since our objective is to find a behavior policy that minimizes the variance in return across multiple GVFs, we use an existing off-policy TD-style variance estimator (Sherstan et al., 2018). This estimator allows us to bootstrap the target values and iteratively update the variance function, making the solution scalable to complex domains.

We define the variance function by $M_\pi^\mu(s)$, which measures the **variance in the return** of target policy $\pi$ in a given state $s$ when actions are sampled under a different behavior policy $\mu$. We describe how to learn this function in Sec. 5. The variance function for a given state and a given state-action pair are defined respectively as:

$$M_\pi^\mu(s) = \text{Var}_\pi(G_t|s_t = s, a \sim \mu) \quad \text{and} \quad M_\pi^\mu(s, a) = \text{Var}_\pi(G_t|s_t = s, a_t = a, a' \sim \mu).$$

**Our objective is to find an optimal behavior policy $\mu^*$ that efficiently collects a single stream of experience to minimize the sum of variances** $M_{\pi_{\{1\ldots N\}}}^\mu$ under some state distribution $d(s)$, as,

$$\mu^* = \arg\min_\mu \sum_{i=1}^N \sum_s d(s) M_{\pi_i}^\mu(s) \quad \text{s.t.} \quad \mu(a|s) \geq 0 \,\&\, \sum_a \mu(a|s) = 1. \tag{2}$$

We solve the above objective function iteratively. At each iteration $k$, `GVFExplorer` produces a behavior policy $\mu_k$. The behavior policy interacts with the environment and gathers data. Using this data, any off-policy TD algorithm can be used to iteratively estimate the variance function $M_\pi^{\mu_k}$. This variance function is plugged into the optimization problem given in Eq. (3) to update to a policy $\mu_{k+1}$ that reduces variance. The iterative procedure is analogous to *policy iteration*, which alternates policy evaluation with policy improvement. Here, we alternate between the variance evaluation and the improvement of behavior policy to minimize the overall sum of variance across all given GVFs.

Here, our aim is to iteratively improve behavior policy and decrease variance functions to estimate the GVF values $V_{\pi_{i=\{1,2,\ldots,N\}}}$ with reducing MSE:

$$\mu_0 \xrightarrow{E} M_{\pi_{i=1,2,\ldots}}^{\mu_0} \xrightarrow{I} \mu_1 \xrightarrow{E} M_{\pi_{i=1,2,\ldots}}^{\mu_1} \ldots \xrightarrow{E} \mu_K,$$

where $\xrightarrow{E}$ denotes *variance estimation* and $\xrightarrow{I}$ denotes *behavior policy improvement*. Next, we present Theorem 4.1 which principally derives the behavior policy update from $\mu_k$ to $\mu_{k+1}$ by solving the objective in Eq. (7). We demonstrate that the behavior policy update in Eq. (3) minimizes the objective by showing that $\mu_{k+1}$ is a better policy than $\mu_k$. The policy $\mu_{k+1}$ is considered as good as, or better than $\mu_k$, if it obtains lesser or equal total variance across all GVFs: $\sum_i M_{\pi_i}^{\mu_{k+1}}(s) \leq \sum_i M_{\pi_i}^{\mu_k}(s)$. The proof of behavior policy improvement is detailed in Theorem 4.2.

## 4.2 Theoretical Solution

**Theorem 4.1.** *(**Behavior Policy Update:**) To find the behavior policy $\mu$ that minimize the variance objective across $N$ given target policies $\{\pi_i\}_{i=1}^N$, we iteratively update $\mu$ by solving the objective in Eq. (2). Given state-action variance function $M_{\pi_i}^{\mu_k}(s, a)$, the solution to Eq. (2) at iteration $k$ is:*

$$\mu_{k+1}(a|s) = \frac{\sqrt{\sum_i \pi_i(a|s)^2 M_{\pi_i}^{\mu_k}(s, a)}}{\sum_{a'} \sqrt{\sum_i \pi_i(a'|s)^2 M_{\pi_i}^{\mu_k}(s, a')}}. \tag{3}$$

*Proof.* The proof is presented in App. A.1. □

Theorem 4.1 describes how to iteratively update the behavior policy $\mu_k$ that minimizes objective in Eq. (2) by using the return variance $M_{\pi_i}^{\mu_k}$. The policy $\mu_{k+1}$ selects actions proportional to their variance, meaning high-variance return $(s, a)$ pairs are explored frequently. By visiting high-variance return pairs, policy gains informative samples and reduce the overall uncertainty. Consequently, this process improves the GVF value predictions and decrease the number of interactions needed for effective learning.

Next Theorem 4.2 ensures that behavior policy $\mu_{k+1}$ either decreases or maintains the total variance across all GVFs relative to $\mu_k$, ensuring consistent progress towards minimizing the variance without oscillation. In simple terms, each policy update ensures the variance does not increase.

**Theorem 4.2.** *(Behavior Policy Improvement:) The behavior policy update in Eq.(3) ensures that the aggregated variances across all target policies $\{\pi_i\}_{i=1}^N$ either decreases or remains unchanged at each iteration $k$. This non-increasing variance property demonstrates that each successive behavior policy $\mu_{k+1}$ is improvement over $\mu_k$ in terms of reducing the total variance. Formally we have,*

$$\sum_{i=1}^N M_{\pi_i}^{\mu_{k+1}}(s) \leq \sum_{i=1}^N M_{\pi_i}^{\mu_k}(s), \forall k, \forall s \in \mathcal{S}.$$

*Proof.* The proof is in App. A.1. $\square$

## 5 Variance Function

The theorems provided in the previous section rely on the variance function $M_{\pi_i}^{\mu_k}$. Here, we study this variance function in detail.

**What is the Variance Function $M$?** In an off-policy context [Sherstan et al. (2018), Jain et al. (2021)], introduced the variance function $M_\pi^\mu$, which estimates the variance in return under a target policy $\pi$ using data from a different behavior policy $\mu$. We will directly use this function $M_\pi^\mu$ as our variance estimator and present it here for completeness. The function $M_\pi^\mu$ for a state-action pair under $\pi$, with an importance sampling correction factor $\rho_t = \frac{\pi(a_t|s_t)}{\mu(a_t|s_t)}$, is defined as:

$$M_\pi^\mu(s,a) = \text{Var}_{a\sim\mu}\left(G_{t,\pi}|s_t = s, a_t = a\right) = \mathbb{E}_{a\sim\mu}\left[\delta_t^2 + \gamma^2 \rho_{t+1}^2 M_\pi^\mu(s_{t+1}, a_{t+1})|s_t = s, a_t = a\right] \tag{4}$$

Here, $G_{t,\pi}$ is the return at time $t$, and $\delta_t = r_t + \gamma\mathbb{E}_{a'\sim\pi}[Q_\pi(s_{t+1}, a')] - Q_\pi(s_t, a_t)$ is the TD error. We use **Expected Sarsa** (Sutton & Barto, 2018) to compute $\delta_t$, eliminating the need for IS by using the expected value of the next state-action pair under $\pi$ for bootstrapping, thus stabilizing the update and lowering the variance. $M_\pi^\mu(s,a)$ relates the variance under $\pi$ from the current state-action pair to the next, where actions are sampled under $\mu$. This allows effective bootstrapping and iterative update using a TD-style method. The state variance function is defined as $M_\pi^\mu(s) = \sum_a \mu(a|s)\rho^2(s,a)M_\pi^\mu(s,a)$.

Note, the true $Q_\pi$ is required to compute the TD error $\delta_t$ in Eq. (4). Following Sherstan et al. (2018), we substitute the value estimate $\hat{Q}$ for the true function $Q_\pi$ to compute $\delta_t$ in Eq. (4). Additionally, we use variance estimates $\hat{M}_\pi^{\mu_k}$ to update the next step policy $\mu_{k+1}$ instead of true variance in Eq. (3). This approach is similar to *generalized policy iteration* (Sutton & Barto, 2018), which alternatively updates the value estimator and then improves the policy.

Next, we prove the existence of $M_\pi^\mu$ in Lemma 5.1, which was not covered in Jain et al. (2021). This proof establishes a loose upper bound on the IS ratio $\rho$, limiting the divergence of the behavior policy $\mu$ from the target policy $\pi$ for effective off-policy variance estimation. This aligns with methods like TRPO (Schulman et al., 2015) and Retrace (Munos et al., 2016), which stabilize policy updates by controlling divergence.

**Lemma 5.1.** *(Variance Function $M$ Existence:) Given a discount factor $0 < \gamma \leq 1$, the variance function $M_\pi^\mu$ exists, if the below condition satisfies, $\mathbb{E}_{a\sim\mu}\left[\rho^2(s,a)\right] < \frac{1}{\gamma^2}$ for all states.*

*Proof.* Proof in App. A.2. $\square$

Note, the optimal $\mu$ for the objective in Eq. (2), might violate the above constraint on $\rho$; we empirically clip $\rho$ to mitigate this problem. Additionally, IS requires $\mu(a|s) = 0$ when $\pi(a|s) = 0$. We empirically ensure $\mu(a|s) > \varepsilon << 1$ for all actions. The same constraint is added for all the baselines for fair comparison.

# 6   Algorithm

We present `GVFExplorer` algorithm, detailed in Algorithm 1. Our approach uses two networks: $Q_\theta$ for value function and $M_w$ for variance, each with $N$ heads (one head for each GVF). Starting with a randomly initialized behavior policy, the agent observes cumulants for $N$ GVFs at each step and updates $Q_\theta$ using off-policy TD. We use **Expected Sarsa** (Sutton & Barto, 2018) for both $Q$ and $M$, eliminating off-policy corrections. The target $Q$ updates follow:

$$Q_{tar}(s_t, a_t, s_{t+1}) = c_t + \gamma \sum_{a'} \pi(a'|s_{t+1}) Q_\theta(s_{t+1}, a'). \tag{5}$$

We use the TD error from $Q$-learning, $\delta_Q = Q_{tar} - Q_\theta$, to update target $M$,

$$M_{tar}(s_t, a_t, s_{t+1}) = \delta_Q^2 + \gamma^2 \sum_{a'} \pi(a'|s_{t+1}) M_w(s_{t+1}, a'). \tag{6}$$

Both networks are updated via an MSE loss. The behavior policy is iteratively updated using the new variance estimates for $K$ steps, with learned $Q$ values used for MSE metrics in Eq. (1).

To ensure reliable estimates, we initialize $M$ values to small non-zero constants and apply epsilon exploration, which decays over time, ensuring coverage of the state-action space. This guarantees that agents visit a broad range of state-action pairs early on, preventing issues of zero variance for unvisited pairs. We applied epsilon-exploration to both `GVFExplorer` and the baselines for fair comparison.

We also use techniques like experience replay Lin (1992) for data reuse and target networks for both $Q$ and $M$ to improve learning stability. Expected Sarsa is used consistently across all baselines for fair comparison. Refer to Algorithm 1 for further details.

---

**Algorithm 1:** `GVFExplorer`: Efficient Behavior Policy Iteration for Multiple GVFs Evaluations

**Input:** Target policies $\pi_{i \in \{1,...n\}}$, initial behavior policy $\mu_1$, replay buffer $\mathcal{D}$, primary networks $Q_\theta, M_w$ (small non-zero $M$), target networks $Q_{\bar\theta}, M_{\bar w}$, learning rates $\alpha_Q, \alpha_M$, mini-batch size $b$, trajectory length $T$, target update frequency $l = 100$, value/variance update frequencies $p = 4$, $m = 8$, training steps $K$, exploration rates $\varepsilon_0, \varepsilon_{\text{decay}}, \varepsilon_{\min}$

1  **for** *environment step* $k = 1, \dots K$ **do**
2  $\quad$ Set exploration rate: $\varepsilon_k = \max(\varepsilon_{\min}, \varepsilon_0 \cdot \varepsilon_{\text{decay}}^k)$
3  $\quad$ Select action $a_t \sim \begin{cases} \mu_k(\cdot|s_t) & \text{if random()} \leq 1 - \varepsilon_k \\ \text{Uniform Policy} & \text{otherwise} \end{cases}$
4  $\quad$ Observe next state $s_{t+1}$ and cumulants $c_t = Vector(size(n))$
5  $\quad$ Store transition $(s_t, a_t, s_{t+1}, c_t)$ in $\mathcal{D}$
6  $\quad$ **if** $k\%p == 0$ **then**
7  $\quad\quad$ //Update the Value $Q_\theta$ network
8  $\quad\quad$ Sample mini-batch of size $b$ of transition $(s_t, a_t, s_{t+1}, c_t) \sim \mathcal{D}$.
9  $\quad\quad$ Update $Q_\theta$ using MSE loss $(Q_{tar}(s_t, a_t) - Q_\theta(s_t, a_t))^2$, where $Q_{tar}$ is Eq. (5).
10 $\quad$ **end**
11 $\quad$ **if** $k\%m == 0$ **then**
12 $\quad\quad$ //Update the Variance $M_w$ network
13 $\quad\quad$ Sample mini-batch of size $b$ of transition $(s_t, a_t, s_{t+1}, c_t) \sim \mathcal{D}$
14 $\quad\quad$ Update $M_w$ using MSE loss $(M_{tar}(s_t, a_t) - M_w(s_t, a_t))^2$, where $M_{tar}$ is Eq. (6).
15 $\quad$ **end**
16 $\quad$ **if** $k\%l == 0$ **then**
17 $\quad\quad$ $\bar w = w$ and $\bar\theta = \theta$ $\quad$ //Update both target networks weights
18 $\quad$ **end**
19 $\quad$ //Update the behavior policy $\mu$ using the new Variance $M_w$
20 $\quad$ Behavior policy becomes: $\mu_{k+1}(a|s) = \dfrac{\sqrt{\sum_{i=1}^n \pi_i(a|s)^2 M_w^i(s,a)}}{\sum_{a' \in \mathcal{A}} \sqrt{\sum_{i=1}^n \pi_i(a'|s)^2 M_w^i(s,a')}}, \forall s \in \mathcal{S}, a \in \mathcal{A}.$
21 **end**
22 **Returns** GVFs Values $V_i(s) = \sum_a \pi_i(a|s) Q_\theta^i(s,a)$ for $i = \{1, \dots, n\}$

---

# 7 Experiments

We investigate the empirical utility of our proposed algorithm in both discrete and continuous state environments. Our experiments are designed to answer the following questions: (a) How does `GVFExplorer` compare with the different baselines (explained below) in terms of convergence speed and estimation quality? (b) Can `GVFExplorer` handle a large number of GVFs evaluations? (c) Can `GVFExplorer` work with non-stationary GVFs which change with time? (d) Can `GVFExplorer` work with non-linear function approximations and complex Mujoco environments? [1]

**Baselines.** We use Off-policy Expected Sarsa updates for parallel GVF estimations for all the experiments (including baselines) for fair comparison. We benchmark against several different **baselines**: (1) `RoundRobin`: uses a round-robin strategy sampling episodically from all target policies (2) `MixturePolicy`: Aggregated policy sampling from all target policies; (3) `SR`: a Successor Representation (SR) method using intrinsic reward of total change in SR and reward weights to learn behavior policy (McLeod et al., 2021). (4) `BPS`: behavior policy search method originally designed for single policy evaluation using a REINFORCE variance estimator (Hanna et al., 2017); we adapted it by averaging variance across multiple GVFs (similar to our objective). `BPS` results are limited to tabular settings due to scalability issues with it. (5) `UniformPolicy`: a uniform sampling policy over the action space. Implementation details and hyperparameters are in App. B.

**Type of Cumulants.** We experiment with three different types of cumulants, similar to McLeod et al. (2021) – **constant** with a fixed value; **distractor**, a *stationary* signal with fixed mean and constant variance (normal distribution); **drifter**, a *non-stationary* cumulant with zero-mean random walk with low variance (vary with time). Further description of cumulants is in App. B.2.

**Experimental Settings.** To answer the questions presented above, we consider different settings: **(Two Distinct Policies & Identical Cumulants):** In a tabular setting, we examine two GVFs with distinct target policies but identical *distractor cumulant*, $(\pi_1, c), (\pi_2, c)$. **(Two Distinct Policies & Distinct Cumulants):** In the same environment, we assess two GVFs with distinct target policy and distinct *distractor cumulant* with different fixed means, $(\pi_1, c_1), (\pi_2, c_2)$. **(Large Scale Evaluation with 40 distinct GVFs):** To verify the scalability of proposed method with high number of GVFs, we evaluate combinations of 4 different target policies $\pi_1 \ldots \pi_4$ with 10 different *constant cumulants* $c_1 \ldots c_{10}$, resulting in 40 GVFs. **(Non-Stationary Cumulants in FourRooms):** In FourRooms environment, we assess with two distinct GVFs - stationary distractor and non-stationary *drifter cumulant* $-$ $(\pi_1, c_1), (\pi_2, c_2)$. **(Non-Linear Function Approximation):** In a continuous state environment with non-linear function approximator, we evaluate two distinct *distractor* GVFs, $(\pi_1, c_1), (\pi_2, c_2)$. **(Mujoco environments):** In Mujoco environments – walker and cheetah – evaluate different GVF tasks like walk, run and flip. Across these varied settings, we measure the averaged MSE across multiple GVFs.

## 7.1 Tabular Experiments

We conducted experiments in $20 \times 20$ gridworld with four cardinal actions and a tabular $20 \times 20$ FourRooms environment for added complexity. The discount factor is $\gamma = 0.99$, and the environment is stochastic with a $0.1$ probability of random movement. The cumulants are zero everywhere except for at the goals. Episode terminates after $500$ steps or upon reaching the goal. True value function for MSE computation is calculated analytically $V_\pi = (I - \gamma P_\pi)^{-1} c_\pi$. Detailed description of target policies and cumulants is provided in App. B.3. Table 1 summarizes the below results for tabular experiments.

In **Two Distinct Policies & Identical Cumulants**, we consider gridworld environment with *distractor* cumulant at top left corner with a reward drawn from normal distribution. Fig. 1a shows the averaged MSE across the two GVFs, with `GVFExplorer` showing much lower MSE compared to baselines.

Next, in **Two Distinct Policies & Distinct Cumulants**, we consider two distinct *distractor* cumulant (with different mean) GVFs placed at top-left and top-right corner respectively. Fig. 2a shows `GVFExplorer` with reduced MSEs compared to baselines. Figs. 2b and 2c qualitatively analyze the average absolute difference between true and estimated GVF values across states, $\mathbb{E}_i[|V_{\pi_i}^{c_i} -$

---

[1]The code is available on Github: https://github.com/arushijain94/GVFExplorer.

$\hat{V}_{\pi_i}^{c_i}|$], showing smaller errors (duller colors) for `GVFExplorer`. Fig. 8 (in App. B.3.2) presents the individual variance and MSE for both GVFs in `GVFExplorer`. Further, we conduct an ablation study to experiment with how `GVFExplorer` performance changes with poorer feature approximations. Fig. 10 (in App. B.3.3) shows that MSE increases as the feature quality deteriorates, but `GVFExplorer` remains robust with moderately coarse approximations.

For **Non-Stationary Cumulant in FourRooms**, we evaluate the performance in FourRooms (FR) environment (Sutton et al., 1999) with two distinct GVFs: stationary **distractor** cumulant and a non-stationary **drifter** cumulant which changes value over time. As shown in Fig. 1b, `GVFExplorer` reduces MSE faster than other baselines, even with the non-stationary cumulant. Fig. 11 (in App. B.3.4) demonstrates the effectiveness of `GVFExplorer` in tracking the non-stationary cumulant signal in the later stages of learning.

In **Large Scale Evaluation with 40 Distinct GVFs**, we evaluate our method's scalability to large number of GVFs (refer App. B.3.5). We use **constant** cumulants with values ranging in $[50, 100]$. Fig. 1c compares the average MSE across the GVFs, showing that `GVFExplorer` scales well with an increasing number of GVFs. In contrast, the `SR` baseline struggles with scalability due to the varying cumulant scales affecting the intrinsic reward (the summation of all SRs and reward weights) of behavior policy.

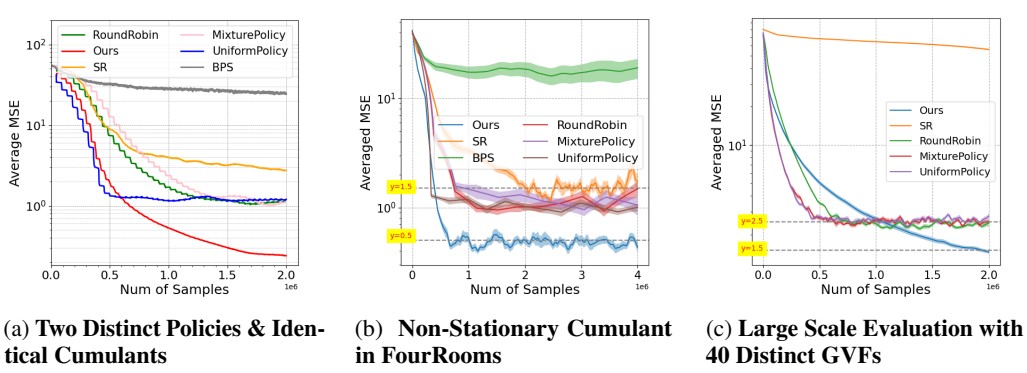

(a) **Two Distinct Policies & Identical Cumulants**

(b) **Non-Stationary Cumulant in FourRooms**

(c) **Large Scale Evaluation with 40 Distinct GVFs**

Figure 1: **MSE Performance**: Averaged MSE over 25 runs with standard error in different experimental settings. `GVFExplorer` demonstrate notably lower MSE compared to the baselines.

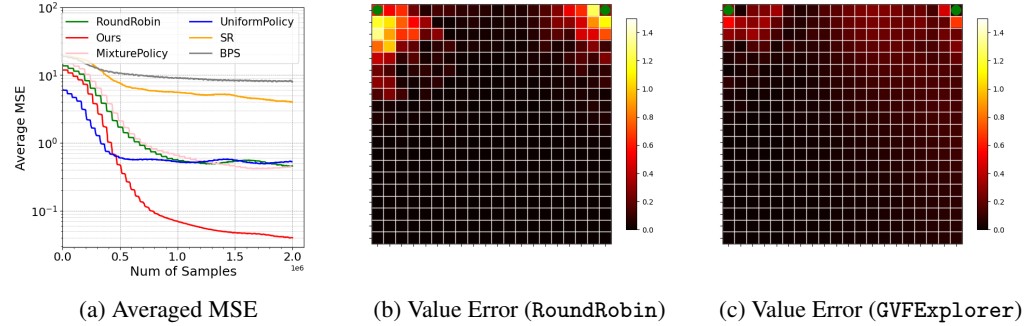

(a) Averaged MSE

(b) Value Error (`RoundRobin`)

(c) Value Error (`GVFExplorer`)

Figure 2: **Two Distinct Policies & Distinct Cumulants**: Evaluate averaged MSE over 25 runs with two distinct distractor GVFs $(\pi_1, c_1), (\pi_2, c_2)$ in gridworld . Green dots at top show two GVF goals. (a) Averaged MSE, (b) averaged absolute error in GVFs value predictions for baseline `RoundRobin` and (c) `GVFExplorer`. *The color bar uses log scale & vibrant colors indicate higher values.*

## 7.2 Continuous State Environment with Non-Linear Function Approximation

We use a continuous grid environment that extends the tabular experiments to a continuous state space (similar to McLeod et al. (2021)) and four discrete actions. For **Non-Linear Function Approximation**, we consider two distinct GVFs with distractor cumulants. An **Experience Replay**

**Buffer** with a capacity of 25K and a batch size of 64 is used for all experiments. Further details on computing true value functions using Monte Carlo and network architectures are in App. B.4.

**Prioritized Experience Replay (PER).** We investigate the integration of PER (Schaul et al., 2015) with our algorithm. Unlike the standard Experience Replay Buffer, which uniformly samples experiences, PER assigns priorities based on the TD error magnitude in the Q-network. PER and `GVFExplorer` are complementary approaches: PER re-weights the collected data in replay buffer based on the priority, while `GVFExplorer` adjusts the behavior policy to influence data collection.

Combining PER with `GVFExplorer` drastically lowers MSE compared to other baselines (even when compared to all baselines + PER). We use the absolute sum of TD errors across multiple GVF Q-functions as a priority metric for PER in all baselines, including `GVFExplorer`. Placing the priority on the TD error of the variance function in `GVFExplorer` yields less favorable results compared to priority on Q-function's TD error. In Fig. 3, we present the MSE for both standard experience replay (solid lines) and PER (dotted lines) for all algorithms. PER generally reduces MSE, but its integration with `GVFExplorer` shows much lower MSE. This is likely as `GVFExplorer` could over-sample high variance return samples, causing a skewed buffer distribution. PER's non-uniform sampling maintains a balanced data distribution, which helps in stringent MSE reduction. For the `SR` baseline, using the TD error in SR predictions as a priority for PER led to performance degradation, suggesting non-stationarity in SRs' TD errors might mislead PER to prioritize less relevant states under the current policy. The original `SR` work by McLeod et al. (2021) does not use PER in the experiments. For PER scenario, we qualitatively compare the absolute value error for baseline `RoundRobin` and `GVFExplorer` by discretizing the state space in Figs. 3b and 3c and observe that our algorithms results in smaller value prediction error. Further insights into the variance estimation by `GVFExplorer` is shown in Figs. 15 and 16 (App. B.4). Table 3((App. B.4) summarizes the results highlighting the performance of various algorithms.

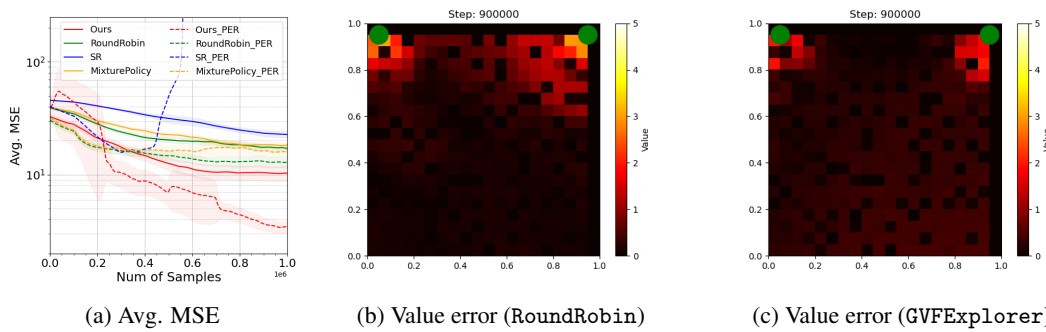

(a) Avg. MSE         (b) Value error (`RoundRobin`)         (c) Value error (`GVFExplorer`)

Figure 3: **Non-Linear Function Approximation**: (a) Averaged MSE over 50 runs with standard error using **Experience Replay Buffer** (solid lines) and **PER** (dotted lines). `GVFExplorer` show lower MSE with both buffers. PER generally reduces MSE across all algorithms except `SR`. Log-scale absolute value error for `RoundRobin` (b) and `GVFExplorer` (c); `GVFExplorer` achieves smaller errors (vibrant colors represent higher values).

## 7.3 Mujoco Environments with Continuous State-Action Tasks

We use DM-Control (Tassa et al., 2018) based continuous state-action tasks to experiment with Mujoco environments, *Walker* and *Cheetah* domain. To expand the proposed method to continuous action environments, any policy-gradient (PG) based algorithm can be used. In our experiments, we use Soft Actor-Critic (SAC) algorithm (Haarnoja et al., 2018) as a base PG method to incorporate the proposed variance-minimization objective.

A separate network for variance estimation is added to SAC. Further implementation details are provided in App. B.5. To experiment in *Walker* environment, we use two GVF tasks, namely 'walk' and 'flip'. Similarly, for *Cheetah* environment, we use 'walk' and 'run' GVF tasks. We also added KL regularize between the learned behavior policy and the given GVFs target policies to prevent divergence. We use MC to compute the true Q-value GVF estimates and compare the MSE between these MC values and the Q-critic network. We use the same Q-critic architecture for the baseline

algorithms – `UniformPolicy` and `RoundRobin` – for fair comparison. In Fig. 4 we observe that
`GVFExplorer` reduces MSE faster than the baselines.

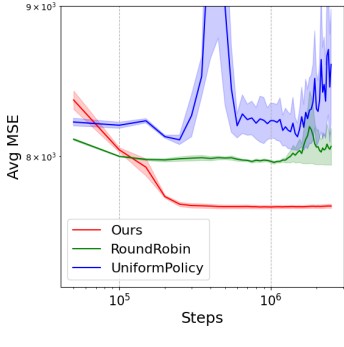

(a) Averaged MSE in Walker

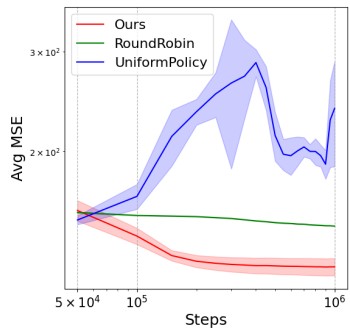

(b) Averaged MSE in Cheetah

Figure 4: **MSE in Mujoco**: Averaged MSE over 5 runs with standard error in Mujoco environment with continuous state-actions for (a)Walker and (b)Cheetah domains for `GVFExplorer`, `UniformPolicy` and `RoundRobin`. `GVFExplorer` consistently lowers averaged MSE as compared to the baselines.

## 8   Conclusion

We addressed the problem of parallel evaluations of multiple GVFs, each conditioned on a given target policy and cumulant. We developed a method to adaptively learn a behavior policy that uses a single experience stream to estimate all GVF values in parallel. The resulting behavior policy update selects the actions in proportion to the total variance of the return across GVFs. This guides the policy to frequently explore less understood areas (high variance in return), which helps to better estimate the mean return with fewer samples. Therefore, our approach lowers the overall MSE in GVF predictions while reducing the number of interactions required. We theoretically proved that each behavior policy update reduces or maintains the total prediction error. Empirically, we showed that `GVFExplorer` scales effectively with an increasing number of distinct GVFs, robustly handles non-stationary cumulants in a tabular setting, and adapts well to non-linear function approximation. Additionally, we showcased its performance in complex continuous state-action Mujoco environments, showing that `GVFExplorer` can be seamlessly integrated with existing policy-gradient methods.

**Limitations and Future Work.**   One notable drawback of `GVFExplorer` is the increased time complexity, due to simultaneously learning two networks for value and variance estimation respectively. Additionally, `GVFExplorer` has not been evaluated in environments with significant difference in the cumulant value range. Such disparities could lead to varying variances, potentially resulting in oversampling areas with higher cumulant values. Calibration across cumulants may be necessary in these cases.

In this work, we focused on minimizing the total MSE, but other loss functions, such as weighted MSE could also be considered. However, weighted MSE requires prior knowledge about the weighting of errors in different GVFs, which is not readily available. A potential future direction could be to use variance scales to automatically adjust these weights to provide uniform MSE reduction across all GVFs. Looking ahead, we are interested in testing our approach with multi-dimensional cumulants and general state-dependent discount factors, as well as, extending the applicability of `GVFExplorer` to control settings where the target policies are unknown.

## Acknowledgments and Disclosure of Funding

We are grateful to the anonymous reviewers for their valuable feedback. We also extend our thanks to Nishanth Anand, Kshitij Jain, Ayush Jain, Pierre-Luc Bacon, and Subhojyoti Mukherjee for their insightful suggestions. Josiah Hanna acknowledges support from NSF (IIS-2410981), American Family Insurance through a research partnership with the University of Wisconsin—Madison's Data Science Institute, and the Wisconsin Alumni Research Foundation.

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

# A Proofs

## A.1 Behavior Policy Update Theorems

**Theorem 4.1.** (**Behavior Policy Update:**) *To find the behavior policy $\mu$ that minimize the variance objective across $N$ given target policies $\{\pi_i\}_{i=1}^N$, we iteratively update $\mu$ by solving the objective in Eq. (2). Given state-action variance function $M_{\pi_i}^{\mu_k}(s, a)$, the solution to Eq. (2) at iteration $k$ is:*

$$\mu_{k+1}(a|s) = \frac{\sqrt{\sum_i \pi_i(a|s)^2 M_{\pi_i}^{\mu_k}(s, a)}}{\sum_{a'} \sqrt{\sum_i \pi_i(a'|s)^2 M_{\pi_i}^{\mu_k}(s, a')}}. \tag{3}$$

*Proof.* We formulate Eq. (2) as a Lagrangian equation below to solve for the optimal behavior policy $\mu^*$.

$$\mathcal{L}(\mu, \lambda_{s,a}, w_s) = \underbrace{\sum_i \sum_s d(s) M_{\pi_i}^\mu(s)}_{\text{I part}} + \underbrace{\sum_{s,a} \lambda_{s,a} \mu(s, a)}_{\text{II part}} + \underbrace{\sum_s w_s (1 - \sum_a \mu(s, a))}_{\text{III part}}. \tag{7}$$

Here, $\lambda \in \mathbb{R}^{|\mathcal{S} \times \mathcal{A}|}$ and $\mathbf{w} \in \mathbb{R}^{|\mathcal{S}|}$ denotes the Lagrangian multipliers. The following KKT conditions satisfy:

1. $\nabla_{\mu(s,a)} \mathcal{L} = 0$
2. $\lambda_{s,a} \mu(s, a) = 0$
3. $\lambda_{s,a} \geq 0$
4. $\mu(s, a) \geq 0$
5. $\sum_a \mu(s, a) = 1$

**Gradient of $\rho$.** The gradient of $\rho(s, a)$ w.r.t. $\mu(a|s)$,

$$\nabla_{\mu(s,a)} \rho(s, a) = \frac{\pi(a|s)}{\nabla \mu(a|s)} = -\frac{\pi(a|s)}{\mu(a|s)^2} = -\frac{\rho(s, a)}{\mu(a|s)}.$$

**Solving I part.** We will compute the gradient of $M_{\pi_i}^\mu(s)$ in Eq. (4) w.r.t to given $\mu(s, a)$. Here, $\rho(s, a) = \frac{\pi(a|s)}{\mu(a|s)}$ is IS weight. We expand $M_{\pi_i}^\mu(s)$ relation with $M_{\pi_i}^\mu(s, a)$ to derive the gradient,

$$M_{\pi_i}^\mu(\tilde{s}) = \sum_{\tilde{a}} \mu(\tilde{a}|\tilde{s}) \rho_i(\tilde{s}, \tilde{a})^2 M_{\pi_i}^\mu(\tilde{s}, \tilde{a})$$

$$\nabla_{\mu(s,a)} M_{\pi_i}^\mu(\tilde{s}) = \nabla_{\mu(s,a)} \left\{ \sum_{\tilde{a}} \mu(\tilde{a}|\tilde{s}) \rho_i(\tilde{s}, \tilde{a})^2 M_{\pi_i}^\mu(\tilde{s}, \tilde{a}) \right\}$$

$$= \rho_i(s, a)^2 M_{\pi_i}^\mu(s, a) + 2\mu(a|s) \rho_i(s, a) \underbrace{\nabla \rho_i(s, a)}_{=-\frac{\rho_i(s, a)}{\mu(a|s)}} M_{\pi_i}^\mu(s, a) + \underbrace{\mu(a|s) \rho_i(s, a)^2 \nabla_\mu M_{\pi_i}^\mu(s, a)}_{=\text{IV part}}$$

$$= \rho_i(s, a)^2 M_{\pi_i}^\mu(s, a) - 2\rho_i(s, a)^2 M_{\pi_i}^\mu(s, a)$$

$$= -\rho_i(s, a)^2 M_{\pi_i}^\mu(s, a).$$

The final term $\nabla_\mu M_{\pi_i}^\mu(s, a)$ is difficult to estimate in an iterative off-policy setting. Hence, we drop the (*IV part*) from the above gradient, which is similar to Degris et al. (2012)[Sec 2.2], where the gradient of $Q(s, a)$ was dropped while deriving the policy update.

Solving for the Lagrangian Eq. (7) further by substituting the (*I part*), and taking derivation of II & III part and using the (1) KKT condition.

$$\nabla_{\mu(s,a)} \mathcal{L}(\mu, \lambda_{s,a}, w_s) = -\sum_i d(s) \rho_i(s, a)^2 M_{\pi_i}^\mu(s, a) + \lambda_{s,a} - w_s = 0. \tag{8}$$

From (2)KKT condition, we know that either $\lambda_{s,a} = 0$ or $\mu(a|s) = 0$. Following the arguments of IS, support for $\mu(a|s)$ can only be 0 when the support for target policy $\pi(a|s) = 0$. Solving for the case when support for target policy in non-zero, then let $\lambda_{s,a} = 0$. We can simplify the gradient of Lagrangian in Eq. (8),

$$w_s = -\sum_i d(s)\rho_i(s,a)^2 M^\mu_{\pi_i}(s,a) = -\sum_i d(s)\frac{\pi_i(a|s)^2}{\mu(a|s)^2}M^\mu_{\pi_i}(s,a)$$

$$\mu(a|s) = \sqrt{\frac{\sum_i \pi_i(a|s)^2 M^\mu_{\pi_i}(s,a)}{-w_s/d(s)}}$$

(9)

We know that the numerator is always positive (variance $M$ is positive), therefore $w_s < 0$. Let $y_s = -w_s/d(s)$. From condition (5), we know that $\sum_a \mu(a|s) = 1$. Using Eq. (9) and summing over all the actions we get,

$$\sum_a \mu(a|s) = \sum_a \sqrt{\frac{\sum_i \pi_i(a|s)^2 M^\mu_{\pi_i}(s,a)}{y_s}} = 1$$

$$\text{Hence, } \sqrt{y_s} = \sum_a \sqrt{\sum_i \pi_i(a|s)^2 M^\mu_{\pi_i}(s,a)}.$$

Therefore, the update for optimal behavior policy becomes,

$$\mu(a|s)^* = \frac{\sqrt{\sum_i \pi_i(a|s)^2 M^{\mu^*}_{\pi_i}(s,a)}}{\sum_a \sqrt{\sum_i \pi_i(a|s)^2 M^{\mu^*}_{\pi_i}(s,a)}}.$$

As the optimal policy $\mu^*$ appear on both the sides, this can be interpreted as an iterative update, where $k$ denotes the iterate number.

$$\mu_{k+1}(a|s) = \frac{\sqrt{\sum_i \pi_i(a|s)^2 M^{\mu_k}_{\pi_i}(s,a)}}{\sum_a \sqrt{\sum_i \pi_i(a|s)^2 M^{\mu_k}_{\pi_i}(s,a)}}.$$

$\square$

**Theorem 4.2.** *(**Behavior Policy Improvement:**) The behavior policy update in Eq.(3) ensures that the aggregated variances across all target policies $\{\pi_i\}_{i=1}^N$ either decreases or remains unchanged at each iteration $k$. This non-increasing variance property demonstrates that each successive behavior policy $\mu_{k+1}$ is improvement over $\mu_k$ in terms of reducing the total variance. Formally we have,*

$$\sum_{i=1}^N M^{\mu_{k+1}}_{\pi_i}(s) \leq \sum_{i=1}^N M^{\mu_k}_{\pi_i}(s), \forall k, \forall s \in \mathcal{S}.$$

*Proof.* Theorem 4.1 suggests, for any given $\mu_k$ behavior policy, the next successive approximation $\mu_{k+1}$ minimizes the objective function Eq. (2), i.e.,

$$\mu_{k+1} = \min_\mu \sum_i \sum_s d(s)\underbrace{M^{\mu_k}_{\pi_i}(s)}_{=I}$$

$$= \min_\mu \sum_i \sum_s d(s)\underbrace{\sum_a \mu(a|s)\frac{\pi_i(a|s)^2}{\mu(a|s)^2}M^{\mu_k}_{\pi_i}(s,a)}_{=M^{\mu_k}_{\pi_i}(s)}.$$

(10)

We will omit writing $d(s)$ and assume that $s \sim d(s)$. Further, we will use the notation $\rho^i_k(s,a) = \frac{\pi_i(a|s)}{\mu_k(a|s)}$ for ease of writing. From Eq. (10), we can establish the relation,

$$\underbrace{\sum_{i,s,a} \mu_k(a|s)\frac{\pi_i(a|s)^2}{\mu_k(a|s)^2}M^{\mu_k}_{\pi_i}(s,a)}_{=M^{\mu_k}_{\pi_i}(s)} \geq \sum_{i,s,a} \mu_{k+1}(a|s)\frac{\pi_i(a|s)^2}{\mu_{k+1}(a|s)^2}M^{\mu_k}_{\pi_i}(s,a).$$

(11)

Now, we will use Eq. (11) relation to further simplify the equation and establish that variance decreases with every update step $k$. We will use the notation $\rho_{t:t+n} = \Pi_{l=0}^{n} \rho_{t+l}$ to denote the products.

$$
\begin{aligned}
\sum_{i,s} M_{\pi_i}^{\mu_k}(s) &\geq \sum_{i,s,a} \mu_{k+1}(a|s) \rho_{k+1}^i(s,a)^2 M_{\pi_i}^{\mu_k}(s,a) \\
&= \sum_{i,s,a} \mu_{k+1}(a|s) \rho_{k+1}^i(s,a)^2 \mathbb{E}_{a\sim\mu_k}[\delta_t^2 + \gamma^2 M_{\pi_i}^{\mu_k}(s_{t+1})|s_t = s] \\
&= \sum_{i,s} \mathbb{E}_{a\sim\mu_{k+1}} \left[ (\rho_t^i)^2 \delta_t^2 + \gamma^2(\rho_t^i)^2 \underbrace{M_{\pi_i}^{\mu_k}(s_{t+1})}_{\text{expand this}} |s_t = s \right] \\
&\geq \sum_{i,s} \mathbb{E}_{a\sim\mu_{k+1}} \left[ (\rho_t^i)^2 \delta_t^2 + \gamma^2(\rho_t^i)^2 \mathbb{E}_{a\sim\mu_{k+1}} \left[ (\rho_{t+1}^i)^2 \delta_{t+1}^2 + \gamma^2(\rho_{t+1}^i)^2 M_{\pi_i}^{\mu_k}(s_{t+2})|s_{t+1} \right] |s_t = s \right] \\
&= \sum_{i,s} \mathbb{E}_{a\sim\mu_{k+1}} \left[ (\rho_t^i)^2 \delta_t^2 + \gamma^2(\rho_t^i)^2(\rho_{t+1}^i)^2 \delta_{t+1}^2 + \gamma^4(\rho_t^i)^2(\rho_{t+1}^i)^2 M_{\pi_i}^{\mu_k}(s_{t+2})|s_t = s \right] \\
&\;\;\vdots \\
&\geq \sum_{i,s} \mathbb{E}_{a\sim\mu_{k+1}} \left[ \rho_{t:t}^2 \delta_t^2 + \gamma^2(\rho_{t:t+1}^i)^2 \delta_{t+1}^2 + \gamma^4(\rho_{t:t+2}^i)^2 \delta_{t+2}^2 + \ldots |s_t = s \right] \\
&\geq \sum_{i,s} M_{\pi_i}^{\mu_{k+1}}(s).
\end{aligned}
$$

$$\tag{12}$$

$\square$

## A.2 When does Variance Function Exists?

Let $\mathbf{c}_\mu \in \mathbb{R}^{|\mathcal{S}\times\mathcal{A}|}$ denote the pseudo-reward $\mathbf{c}_\mu(s,a) = \sum_{s'} P(s'|s,a)\delta^2(s,a,s')$ and $\bar{\mathbf{P}}_\mu \in \mathbb{R}^{|\mathcal{S}\times\mathcal{A}\times\mathcal{S}\times\mathcal{A}|}$ represent the transition probability matrix $\bar{\mathbf{P}}_\mu(s,a,s',a') = P(s'|s,a)\mu(a'|s')\rho^2(s',a')$. The matrix form of $M_\pi^\mu$ is:

$$M_\pi^\mu = \mathbf{c}_\mu + \gamma^2 \bar{\mathbf{P}}_\mu M_\pi^\mu \implies M_\pi^\mu = (I - \gamma^2 \bar{\mathbf{P}}_\mu)^{-1} \mathbf{c}_\mu. \tag{13}$$

The existence of $M_\pi^\mu$ hinges on the invertibility of matrix $(I - \gamma^2 \bar{\mathbf{P}}_\mu)$. Lemma 5.1 establishes the existence of the above inverse using Definition A.1 and Lemmas A.2 and A.3.

**Definition A.1. (Spectral Radius)** The spectral radius of a matrix $\mathbf{A} \in \mathbb{R}^{n\times n}$ is denoted by $sr(\mathbf{A}) = \max(\lambda_1, \lambda_2, \ldots, \lambda_n)$, where $\lambda_i$ denotes the $i^{th}$ eigenvalue of $\mathbf{A}$.

**Lemma A.2.** *The spectral radius $sr(\mathbf{A})$ of a matrix $\mathbf{A} \in \mathbb{R}^{n\times n}$ follows the relation, $sr(\mathbf{A}) \leq \|\mathbf{A}\|$, where, $\|\mathbf{A}\| = \max_i \sum_j \mathbf{A}(i,j)$ is the infinity norm over a matrix.*

*Proof.* Following the derivation from Bacon (2018) Ph.D. thesis and work of Watkins (2004), we use the eigenvalue of a matrix to show that $sr(\mathbf{A}) < \|\mathbf{A}\|$. We can write $\lambda\mathbf{x} = \mathbf{A}\mathbf{x}$, when $\lambda$ is the eigenvalue of $\mathbf{A}$. For any sub-multiplicative matrix norm, $\|\mathbf{A}\mathbf{B}\| \leq \|\mathbf{A}\|\|\mathbf{B}\|$. Using this property,

$$\|\lambda\mathbf{x}\| = |\lambda|\|\mathbf{x}\| = \|\mathbf{A}\mathbf{x}\| \leq \|\mathbf{A}\|\|\mathbf{x}\|,$$
$$|\lambda| \leq \|\mathbf{A}\|.$$

The above is true for any eigenvalue $\lambda$ of $\mathbf{A}$. So this must also be true for the maximum eigenvalue of $\mathbf{A}$. Therefore, we can express,

$$sr(\mathbf{A}) \leq \|A\|.$$

$\square$

**Lemma A.3.** *When the spectral radius of $sr(\mathbf{A}) < 1$, then $(I - \mathbf{A})^{-1}$ exits and is equal to, $(I - \mathbf{A})^{-1} = \sum_{t=0}^{\infty} \mathbf{A}^t$.*

*Proof.* The proof for the Lemma is presented in Puterman (2014)[Proposition A.3]. $\square$

**Lemma 5.1.** *(Variance Function $M$ **Existence:**) Given a discount factor $0 < \gamma \leq 1$, the variance function $M_\pi^\mu$ exists, if the below condition satisfies, $\mathbb{E}_{a \sim \mu}\left[\rho^2(s,a)\right] < \frac{1}{\gamma^2}$ for all states.*

*Proof.* Following Lemma A.3, the existence of $M$ hinges on the existence of inverse $(I - \gamma^2 \bar{\mathbf{P}}_\mu)^{-1}$. Further, $(I - \gamma^2 \bar{\mathbf{P}}_\mu)^{-1}$ exists if spectral radius $sr(\gamma^2 \bar{\mathbf{P}}_\mu) < 1$. Further, from Lemma A.2, we know that for any given matrix $\mathbf{A}$, spectral radius satisfies, $sr(\mathbf{A}) \leq \|A\|$. Hence, using the above two lemmas, we can express,

$$sr(\gamma^2 \bar{\mathbf{P}}_\mu) \leq \|\gamma^2 \bar{\mathbf{P}}_\mu\| \leq \gamma^2 \|\bar{\mathbf{P}}_\mu\|.$$

Further, if spectral radius satisfies the below condition, then the inverse exists,

$$sr(\gamma^2 \bar{\mathbf{P}}_\mu) \leq \gamma^2 \|\bar{\mathbf{P}}_\mu\| < 1.$$

We expand the middle infinity norm term and get

$$\max_{s,a} \sum_{s',a'} \bar{\mathbf{P}}_\mu(s,a,s',a') < \frac{1}{\gamma^2}$$

$$\max_{s,a} \sum_{s'} P(s'|s,a) \sum_{a'} \mu(a'|s') \rho^2(s',a') < \frac{1}{\gamma^2}.$$

We can further express the above condition as $\mathbb{E}_{a \sim \mu}\left[\rho(s,a)^2\right] < \frac{1}{\gamma^2}, \forall s \in \mathcal{S}$. $\qquad\square$

# B  Experiments

This section provides detail about the experiments in the main paper as well as additional experiments in the Appendix. All the experiments require less than $1GB$ of memory and have used combined compute less than total 4 CPU months and 1 GPU month.

For all the experiments, we consider the two target policies with four cardinal directions left (L), right (R), up (U) and down (D) for the tabular and non-linear function approximation environments. These policies are specified as follows for every state $s \in \mathcal{S}$:

$$\pi_1(s) = \{L : 0.175, R : 0.175, U : 0.25, D : 0.4\}$$
$$\pi_2(s) = \{L : 0.25, R : 0.15, U : 0.25, D : 0.35\}. \tag{14}$$

## B.1  Baselines

(1) `RoundRobin`: We used a round robin strategy to sample data from all given $n$ target policies episodically. We used Expected Sarsa to estimate all GVF value functions in parallel when a transition is given as $(s_t, a_t, s_{t+1}, c_{i=\{1,...n\}})$,

$$Q_i(s_t, a_t) = Q_i(s_t, a_t) + \alpha \left( c_i(s_t, a_t) + \gamma \sum_a \pi_i(a'|s_{t+1}) Q_i(s_{t+1}, a') - Q_i(s_t, a_t) \right)$$

(2) `MixturePolicy`, `UniformPolicy` are also evaluated using Expected Sarsa. `MixturePolicy` is defined as,

$$\mu_{\text{Mixture}}(a|s) = \frac{\sum_{i=1}^N \pi_i(a|s)}{\sum_{a'} \sum_{i=1}^N \pi_i(a'|s)}.$$

(3) `SR`: Based on (McLeod et al., 2021), `SR` uses the summation of weight changes in the Successor Representation (SR) and reward weights to obtain the intrinsic reward for behavior policy updates. We use Expected Sarsa to learn both the SR and the Q-value function from the intrinsic reward. The behavior policy is generated using a Boltzmann policy over the learned Q function, as it empirically performs better than a greedy policy. We apply simple TD Expected Sarsa updates instead of Emphatic TD($\lambda$) as shown in Algo 2 in McLeod et al. (2021). The learning rates for SR, reward weights, and behavior policy Q function are kept the same.

(4) `BPS`: (Hanna et al., 2017) Use a Reinforce style estimator to learn $IS(\tau, \pi) = G(\tau) \Pi_{t=1}^T \frac{\pi(a_t|s_t)}{\mu(a_t|s_t)}$, as mentioned in the original paper. Since, the original work is only about single policy evaluation,

we extended for multiple GVFs by updating behavior policy as summation over $\sum_i IS(\tau, \pi_i)$. The behavior policy weight $\theta$ is updated as:

$$\theta_\mu = \theta_\mu + \alpha \sum_{i=1}^{n} IS(\tau, \pi_i)^2 \sum_{t=1}^{T} \nabla_\theta \log \mu_\theta(a_t|s_t).$$

## B.2 Types of Cumulants

We consider three different types of cumulants similar to McLeod et al. (2021) as shown in Fig. 5:

- **Constant:** Fixed value cumulant, $c_i^t = c_i$

- **Distractor:** *Stationary cumulant* with reward drawn from normal distribution with fixed mean and variance, $c_i^t = N(\mu_i, \sigma_i)$

- **Drifter:** *Non-Stationary cumulant* whose value change over time, $c_i^t = c_i^{t-1} + N(\mu_i, \sigma_i), c_i^0 = 100$.

## B.3 Tabular Experiments

We consider a tabular $20 \times 20$ grid environment with stochastic dynamics. We use the above target policies for the different experimental settings. The Table 1 summarizes the averaged MSE with the same $2 \times 10^6$ samples for different experimental settings.

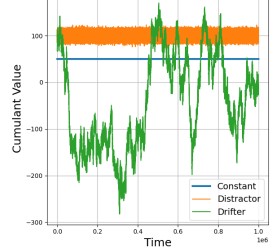

Figure 5: Visual representation of cumulants.

Table 1: **Avg. MSE Summary in Tabular Env:** Compares the average MSE across multiple GVFs in different experimental settings in Tabular environment. We compare baselines with `GVFExplorer` at same $2 \times 10^6$ steps of learning. We show the % improvement in `GVFExplorer` w.r.t. to best baseline `RoundRobin`. *Note: Smaller MSE indicates better performance.*

| Avg MSE @ $2e6$ **steps** | BPS | SR | UniformPol | RoundRobin | MixPol | GVFExplorer (Ours) | % Improvement of Ours (against best baseline) |
|---|---|---|---|---|---|---|---|
| **Distinct policies same cumulant** | 26.13 | 2.76 | 1.22 | 1.15 | 1.15 | **0.24** | 79% |
| **Distinct policies distinct cumulants** | 8.2 | 4.1 | 0.54 | 0.47 | 0.44 | **0.04** | 91% |
| **Non-Stationary cumulants in FR env ($4M$ steps)** | 18.57 | 1.78 | 0.9 | 1.28 | 1.08 | **0.46** | 48% |
| **Large num of GVFs** | - | 53.3 | 2.66 | 2.35 | 2.64 | **1.66** | 29% |

Table 2: **Optimized Hyperparameters:** We show the optimized hyperparameters for different Experimental Settings. $\alpha_Q$ is learning rate for value function. $\alpha_M$ is learning rate for variance function.

| Exp. Settings | distinct policies identical cumulants | distinct policies distinct cumulants | large scale 40 GVF eval | non-linear func approx | non-stationary cumulant in FR |
|---|---|---|---|---|---|
| GVFExplorer (Ours) | ($\alpha_Q = 0.25$, $\alpha_M = 0.8$) | ($\alpha_Q = 0.1$, $\alpha_M = 0.8$) | ($\alpha_Q = 0.5$, $\alpha_M = 0.95$) | ($\alpha_Q = 5e-3$, $\alpha_M = 5e-3$) | ($\alpha_Q = 0.5$, $\alpha_M = 0.8$) |
| RoundRobin | 0.95 | 0.8 | 0.8 | $5e-4$ | 0.8 |
| MixturePolicy | 0.95 | 0.8 | 0.8 | $5e-4$ | 0.8 |
| UniformPolicy | 0.95 | 0.8 | 0.8 | - | 0.8 |
| SR | 0.25 | 0.5 | 0.25 | $1e-3$ | 0.8 |
| BPS | 0.5 | 0.8 | - | - | 0.8 |

**Hyperparameter Tuning:** In our experiments, we use linearly decaying learning rates that starts with initial value of $1.0$ and gradually decreased to an optimized minimum value within $500K$ steps of environmental interactions. We used different learning rates for value and variance function in `GVFExplorer`. The minimum learning rate parameter was swept within $\{0.1, 0.25, 0.5, 0.8, 0.9, 0.95\}$ for both value and variance function. The optimal minimum learning rate was determined based on the one achieving the lowest average Mean Squared Error

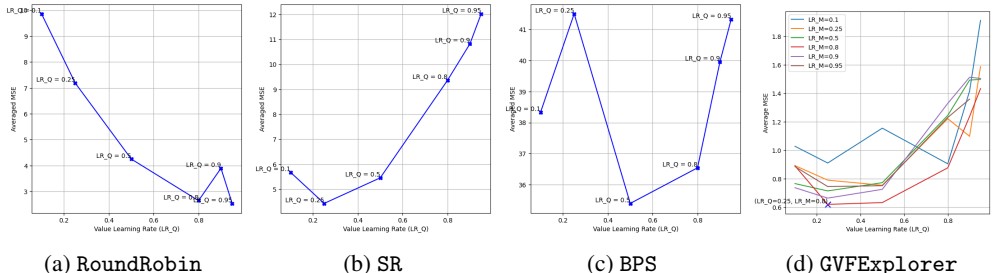

| (a) RoundRobin | (b) SR | (c) BPS | (d) GVFExplorer |

Figure 6: **Impact of Learning Rate on Averaged MSE in Two Distinct Policies & Identical Cumulants scenario**: Demonstrate the effect of changing minimum value of learning rate on the averaged MSE (performance averaged over 10 runs) across GVFs. The optimal hyperparameter is selected based on the least MSE in these plots. LR_Q: value learning rate, LR_M: variance learning rate.

(MSE) after $800K$ sample interactions. **This hyperparameter tuning approach was consistently applied for all algorithms including baselines**. Fig. 6 shows the sensitivity analysis of varying learning rates for value functions (all baselines) and variance functions (our method) with the averaged MSE performance in **Two Distinct Policies & Identical Cumulants**. The learning rate resulting in the lowest MSE was selected as optimal. In Table 2, we show the optimal hyperparameters for the four experimental settings discussed in the paper earlier (refer *Experimental Settings* in Sec. 7).

### B.3.1 Two Distinct Policies & Identical Cumulants

In tabular $20 \times 20$ grid, we consider distinct target policies with identical distractor cumulant $r = N(\mu = 100, \sigma = 5)$. In Fig. 7, we depict the individual MSE over 25 runs for both the GVFs $(\pi_1, c), (\pi_2, c)$; showing decreased MSE for GVFExplorer compared to the baselines.

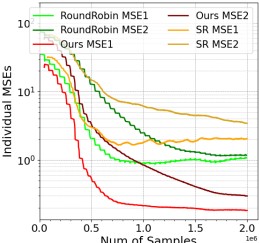

Figure 7: **Two Distinct Policies & Identical Cumulants in 20x20 Grid**: Averaged MSE over 25 runs for two GVFs $(\pi_1, c), (\pi_2, c)$ with same cumulant. We show individual $\text{MSE}_1, \text{MSE}_2$. GVFExplorer shows lower MSE compared to other baselines.

### B.3.2 Two Distinct Policies & Distinct Cumulants

In tabular $20 \times 20$ grid, we consider distinct target policies and distinct distractor cumulants $c_1 = N(\mu = 100, \sigma = 5)$ placed at top-left corner and $c_2 = N(\mu = 50, \sigma = 5)$ placed in top-right corner. Fig. 8 shows the individual MSE for all algorithms and the estimated variance in GVFExplorer for both the GVFs.

**Semi-greedy $\pi$ for Two Distinct Policies & Distinct Cumulants:** We evaluated semi-greedy target policies with distinct cumulants, $(\pi_1, c_1)$ and $(\pi_2, c_2)$ within a 20x20 grid. The target policies are designed with a bias towards top-left and top-right goals respectively,

$$\pi_1(s) = \{L : 0.4, R : 0.1, U : 0.4, D : 0.1\} \forall s \in \mathcal{S}$$
$$\pi_2(s) = \{L : 0.1, R : 0.4, U : 0.4, D : 0.1\} \forall s \in \mathcal{S}. \tag{15}$$

We keep the same cumulants same as in previous experiment. Fig. 9 compares the average MSE performance, where GVFExplorer exhibits comparable MSE to RoundRobin baseline but requires

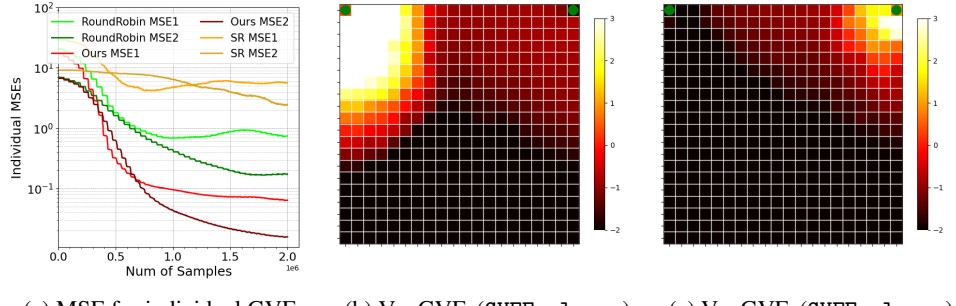

(a) MSE for individual GVFs   (b) Var $\text{GVF}_1$(`GVFExplorer`)   (c) Var $\text{GVF}_2$(`GVFExplorer`)

Figure 8: **Two Distinct Policies & Distinct Cumulants in Grid env**: Evaluate two distinct GVFs $(\pi_1, c_1)$ and $(\pi_2, c_2)$ averaged over 25 runs. Compared baselines – `RoundRobin`, `MixturePolicy`, `UniformPolicy`, SR, BPS with `GVFExplorer`. Green dots show GVF goals. (a) Individual $\text{MSE}_1$ for GVF $(\pi_1, c_1)$, and $\text{MSE}_2$ for GVF $(\pi_2, c_2)$. (b,c) Estimated variance $\hat{M}_{\pi_1}^{c_1}, \hat{M}_{\pi_2}^{c_2}$ in `GVFExplorer`. Variance plots show log-scale empirical values; most areas appear black, due to their relatively small magnitude compared to high variance regions. *The color bar uses log scale & vibrant colors indicate higher values.*

more samples to converge. This outcome can be attributed to the near-greedy nature of the target policies, which inherently guides `RoundRobin` along goal directed trajectories, enabling it to achieve nearly accurate predictions with fewer samples. The optimal hyperparameters for `RoundRobin`, `UniformPolicy` and `MixturePolicy` is $\alpha_Q = 0.95$. We used $\alpha_Q = 0.5, \alpha_M = 0.8$ for ours `GVFExplorer`. Another baseline SR has $\alpha_Q = 0.8$ and BPS as $\alpha_Q = 0.9$ as optimal hyperparameters.

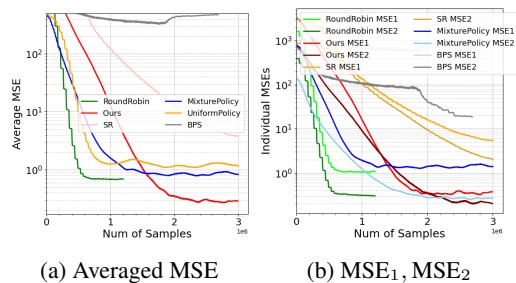

(a) Averaged MSE   (b) $\text{MSE}_1, \text{MSE}_2$

Figure 9: **Semi-greedy target policies in Two Distinct Policies & Distinct Cumulants**: Analysis of MSE averaged over 25 runs with semi-greedy target policies. (a) Averaged MSE (b) $\text{MSE}_1, \text{MSE}_2$. We observe a slower convergence of `GVFExplorer` as compared to baselines like `RoundRobin`, `MixturePolicy` due to target policies being semi-greedy.

### B.3.3   Ablation Study on Effect in Performance on using Poor Feature Approximator

In this section, we study the effect of using degraded approximations or feature quality on the performance metrics. We conducted an ablation study in a 20x20 grid with two distinct *distractor* GVFs with cumulants, $c_1 = N(\mu = 100, \sigma = 5)$ placed on the top-left corner and $c_2 = N(\mu = 50, \sigma = 5)$ on the top-right corner. We reduced the state space into 10x20 and 5x20 feature grids (grouping factors of 2 and 4, respectively), and compared results against the original setup (no approximation, factor = 1). As shown in Fig. 10, the MSE increases as the feature quality deteriorates. Despite this, `GVFExplorer` outperforms `RoundRobin` and `MixturePolicy`, though with very poor approximations (factor = 4), the `UniformPolicy` performs better due to inaccurate variance estimates. These results demonstrate that `GVFExplorer` is robust with moderately coarse approximations but can degrade with significantly poor feature representations, as expected. Further, these results are strengthened by the performance of `GVFExplorer` in Mujoco environments Sec. 7.3.

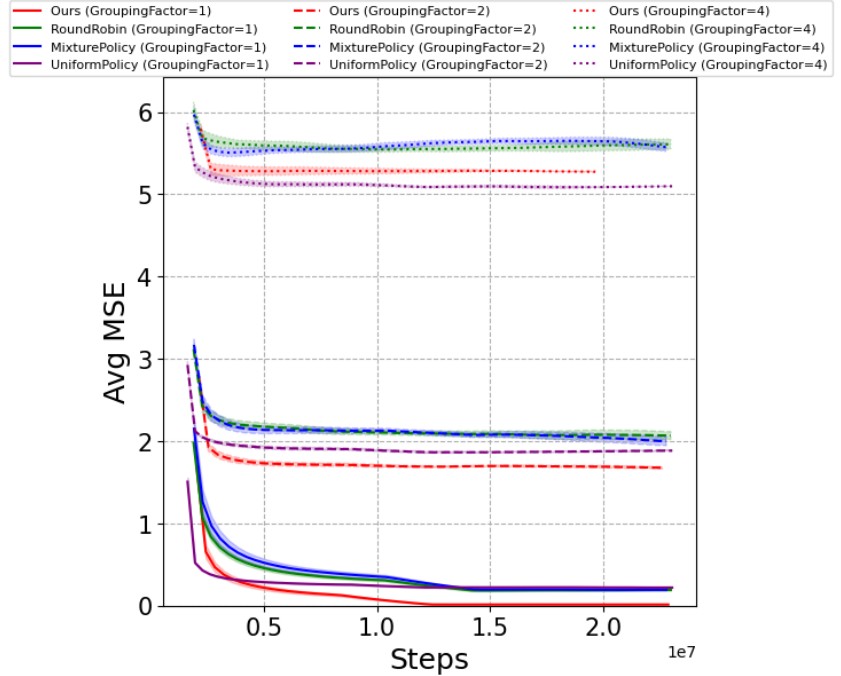

Figure 10: **Impact of Feature Approximation on MSE:** Averaged MSE over 10 runs with standard error in tabular environment. Increasing *GroupingFactor* indicates coarser feature mapping (more states mapped to the same feature). As expected, overall MSE increases with coarser mapping. `GVFExplorer` outperforms baselines given a reasonable feature approximator.

### B.3.4   Non-Stationary Cumulant in FourRooms

In complex FourRooms environment, we consider two distinct target policies in Eq. (14) and different cumulants – stationary **distractor** with $N(\mu = 100, \sigma = 2)$ in top-left room, non-stationary **drifter** signal of $\sigma = 0.5$ in top-right room. Figs. 11a and 11b shows the change in estimated variance $M$ of `GVFExplorer` from early learning steps to later steps (vibrant color shows higher numerical value). We experimented with changing the value of $\sigma$ of drifter cumulant to see the effect on MSE. In Fig. 11c we observe that MSE increases with increasing the value of driftness ($\sigma$) over time.

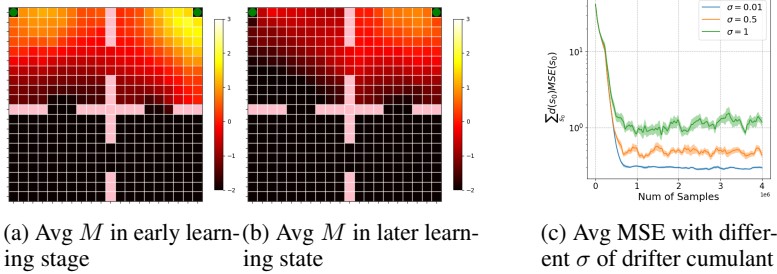

(a) Avg $M$ in early learning stage  (b) Avg $M$ in later learning state  (c) Avg MSE with different $\sigma$ of drifter cumulant

Figure 11: **Non-Stationary Cumulant in FourRooms**: We placed a stationary distractor cumulant in the top-left room and a non-stationary drifter cumulant in the top-right room. (a & b) show the change in estimated variance $\hat{M}$ over time, highlighting the effectiveness of `GVFExplorer` in tracking the non-stationary cumulant placed in top-right corner, later in learning process over stationary cumulant (top-left). (c) shows the average MSE for `GVFExplorer` with different levels of driftness ($\sigma$) in the cumulant value; higher driftness leads to higher MSE.

### B.3.5 Large Scale Evaluation with 40 Distinct GVFs

We evaluate `GVFExplorer` ability to handle a large number of GVFs. We examine four target policies ($\pi_{n\in 1...4}$), each aligned with a cardinal direction, and ten cumulants ($c_{m\in 1...10}$), aiming to predict 40 GVF combinations ($v_{\pi_{1...4}}^{c_1} \ldots v_{\pi_{1...4}}^{c_{10}}$). Each GVF is associated with a state space region ("goal"), uniformly sampled and assigned **constant** cumulant value ranging $[50, 100]$, demonstrated in Fig. 12. In this setting, we considered 4 target policies in the four cardinal directions, namely:

$$\pi_N(s) = \{L : 0.1, R : 0.1, U : 0.7, D : 0.1\}\forall s \in \mathcal{S}$$
$$\pi_E(s) = \{L : 0.1, R : 0.7, U : 0.1, D : 0.1\}\forall s \in \mathcal{S}$$
$$\pi_S(s) = \{L : 0.1, R : 0.1, U : 0.1, D : 0.7\}\forall s \in \mathcal{S}$$
$$\pi_W(s) = \{L : 0.7, R : 0.1, U : 0.1, D : 0.1\}\forall s \in \mathcal{S}.$$

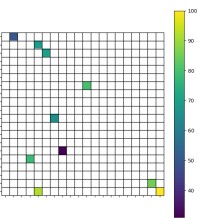

Figure 12: 10 different cumulants for **Large Scale Evaluation with 40 Distinct GVFs** in $20 \times 20$ grid. The color depict the cumulant empirical value.

### B.3.6 Ablation: IS ratios vs Expected Sarsa Update

In FourRooms environment with distractor and drifter cumulants (two distinct GVFs), we compare the following off-policy update styles: (1) Off-policy TD updates using IS $\rho$ correction,

$$Q(s_t, a_t) = Q(s_t, a_t) + \alpha_Q \underbrace{(c_t + \gamma \rho_{t+1} Q(s_{t+1}, a_{t+1}) - Q(s_t, a_t))}_{=\delta_Q}$$

$$M(s_t, a_t) = M(s_t, a_t) + \alpha_M (\delta_Q^2 + \gamma^2 \rho_{t+1}^2 M(s_{t+1}, a_{t+1}) - M(s_t, a_t))$$

and (2) Expected Sarsa update in Eqs. (5) and (6). In Fig. 13, we observe that Expected Sarsa leads to lower MSE, hence we use Expected Sarsa for all the TD updates in *all the algorithms* including baseline for further update stability.

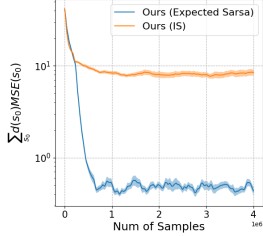

Figure 13: **IS vs Expected Sarsa update in FR env:** We show the averaged MSE (over 25 runs) in Fourrooms by doing off-policy IS corrections in TD updates and off-policy Expected Sarsa in `GVFExplorer` algorithm. Expected Sarsa leads to smaller MSE and faster convergence.

## B.4 Continuous State Environment with Non-linear Function Approximation

**Continuous Environment.** We extend the tabular GridWorld environment to a continuous state space, similar to the approach by McLeod et al. (2021). The environment is a square of dimension $1 \times 1$ with four discrete actions. We evaluate two GVFs: the first GVF has a cumulant at the top-left corner, $c_1 = \mathcal{N}(\mu = 100, \sigma = 5)$, and the second at the top-right corner, $c_2 = \mathcal{N}(\mu = 50, \sigma = 5)$. The target policies are consistent with those used in the tabular environment. The agent receives a

zero cumulant signal elsewhere and moves $0.025$ units in the selected direction with added uniform noise $\mathcal{U}[-0.01, 0.01]$. Episodes start randomly, excluding a $0.05$ radius from the goal state, and end after 500 steps or upon reaching a $0.05$ radius from the goal.

Fig. 14 presents the individual MSE for both GVFs under standard Experience Replay and PER. Our method, `GVFExplorer`, consistently achieves lower MSE compared to baselines. Fig. 15 shows the absolute GVF value prediction error with PER for both the baseline `RoundRobin` and `GVFExplorer`. Fig. 16 illustrates the estimated variance from each GVF, underscoring the necessity for a sampling strategy that prioritizes high-variance return areas to reduce data interactions and ultimately reducing the variance and MSE. Fig. 17 depicts the trajectories sampled from baseline `RoundRobin` and `GVFExplorer`. Table 3 summarizes the performance of various algorithms in this continuous environment.

**Computation of True GVF Values.** The true GVF values in a continuous environment are computed using a Monte Carlo (MC) method. The continuous state space is discretized into a grid, with an initial state sampled from each grid cell. We calculate the average discounted return over $200,000$ trajectories following policy $\pi_i$ with cumulant $c_i$. The mean squared error (MSE) between the estimated and true GVF values is then computed using these discretized states, expressed as $\mathbb{E}_i \left[ \sum_s \left( V_{\pi_i}^{c_i}(s) - \hat{V}_{\pi_i}^{c_i}(s) \right)^2 \right]$ for all algorithms.

**Network Architecture.** We use distinct deep networks for learning value $Q$ and variance $M$. Both networks share a similar architecture, with a shared feature extractor for input states and separate output heads for each GVF, producing multidimensional outputs for both value and variance. The variance network includes a Softplus layer before each head's output to ensure positive numerical values.

Table 3: **Avg. MSE Summary for Continuous Env.**: Averaged MSE across two GVFs for different algorithms in the continuous environment. `GVFExplorer` performance measured against others using **standard** and **prioritized experience replay** after $1 \times 10^6$ learning steps. *Note: Smaller MSE indicates better performance.*

| Avg MSE @1e6 steps | SR | MixturePolicy | RoundRobin | GVFExplorer (Ours) | % Improvement of GVFExplorer (against best baseline) |
|---|---|---|---|---|---|
| **Standard Replay Buffer** | 21.7 | 18.25 | 16.78 | **5.19** | 69% |
| **Prioritized Exp. Replay** | 112 | 14.7 | 11.62 | **3.87** | 66% |

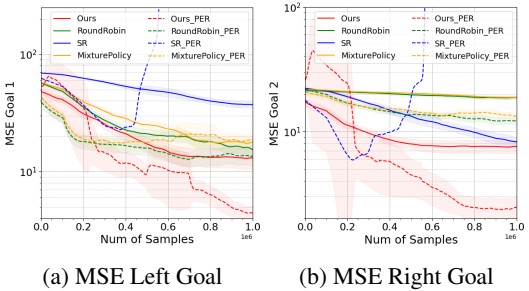

(a) MSE Left Goal      (b) MSE Right Goal

Figure 14: **Individual MSE in Continuous Env.**: Compare the MSE metrics in baselines - `RoundRobin`, `MixturePolicy`, `SR` and `GVFExplorer` (averaged over 50 runs with standard errors) for both standard **Experience Replay Buffer** (solid lines) and with **Priority Experience Replay** (PER) (dotted lines). `GVFExplorer` demonstrates lower MSE with both types of replay buffers. PER generally reduces MSE across all algorithms, except for `SR`.

## B.5 Mujoco Environment with Continuous State-Action Tasks

We conducted additional experiments using the DM-Control suite in the Mujoco environment, focusing on the *Walker* and *Cheetah* domains. For the *Walker* domain, we defined two distinct

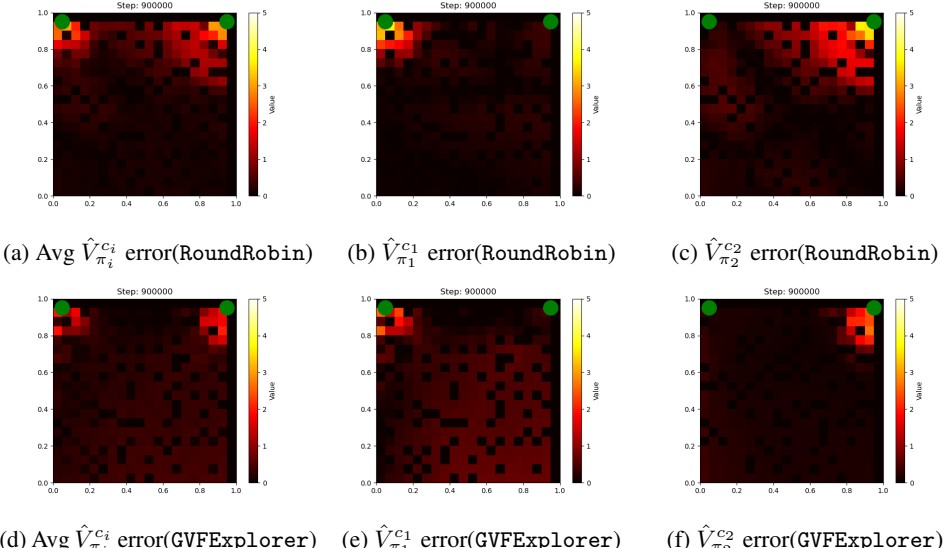

(a) Avg $\hat{V}_{\pi_i}^{c_i}$ error(RoundRobin)  (b) $\hat{V}_{\pi_1}^{c_1}$ error(RoundRobin)  (c) $\hat{V}_{\pi_2}^{c_2}$ error(RoundRobin)

(d) Avg $\hat{V}_{\pi_i}^{c_i}$ error(GVFExplorer)  (e) $\hat{V}_{\pi_1}^{c_1}$ error(GVFExplorer)  (f) $\hat{V}_{\pi_2}^{c_2}$ error(GVFExplorer)

Figure 15: **Value Prediction Errors in Continuous Env**: Compares log-scale absolute errors between actual and predicted values for two GVFs. Top row: RoundRobin baseline errors; Bottom row: GVFExplorer results at equivalent steps. **(Col 1)**: Mean error, **(Col 2)**: Error in GVF 1, **(Col 3)**: Error in GVF 2. GVFExplorer specially achieves smaller errors in areas where RoundRobin has higher MSE, due to the focus on reducing overall MSE (indicated by lighter colors).

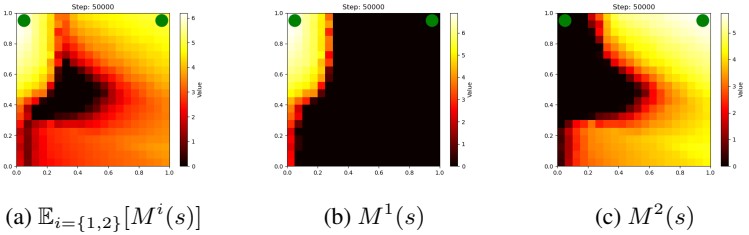

(a) $\mathbb{E}_{i=\{1,2\}}[M^i(s)]$  (b) $M^1(s)$  (c) $M^2(s)$

Figure 16: **Estimated Variance in Continuous Env**: The two GVF goals are depicted in Green. We show the estimated variance $M$ (log values) over states from GVFExplorer method highlighting the motivation for behavior policy to visit high variance areas. (a) Mean variance, (b) Variance for left goal GVF, (c) variance for right goal GVF. These variance plots show log scale empirical values; most areas appear black due to their relatively small magnitude compared to high variance regions.

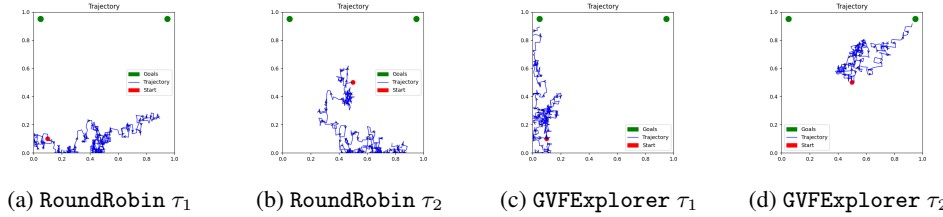

(a) RoundRobin $\tau_1$  (b) RoundRobin $\tau_2$  (c) GVFExplorer $\tau_1$  (d) GVFExplorer $\tau_2$

Figure 17: **Sampled trajectories in Continuous Env**: GVFExplorer generates trajectories which reduces the overall variance, thus minimizing the total MSE. Contrary, RoundRobin collects data according to given target policies. Green dots show GVF goals and red depicts the start state.

GVFs: *walk* and *flip*. In the *Cheetah* domain, we evaluated two GVFs: *walk* and *run*. To handle the continuous action space in these environments, we leveraged policy gradient methods, which are essential when working with continuous actions, as value-based methods like Q-learning are not directly applicable.

Our method can be incorporated with any policy gradient (PG) algorithm. For these experiments, we used Soft Actor-Critic (SAC)(Haarnoja et al., 2018) which provides stability in such settings. SAC uses an entropy regularizer to encourage **exploration**, where the regularization coefficient $\alpha$ is learned adaptively. This allows the agent to balance exploration and exploitation effectively. We present the SAC-`GVFExplorer` in Algorithm 2, a modified version of SAC designed to efficiently handle parallel estimation of multiple GVFs, each aligned with a specific target policy and cumulant.

The first modification involves incorporating a separate variance network, $M_{w_d}(s, a)$, which estimates the variance of returns and supports the behavior policy's objective to reduce mean squared error (MSE) in GVF estimation. This variance network, parameterized by weights $w_1$ and $w_2$, updates alongside the Q-value critics to capture return variability more effectively.

The Q-value target is calculated like in standard SAC algorithm, which uses double-Q trick and entropy term for enhanced exploration. A slight difference from standard SAC is sampling the next state action $a'$ from the target policy $\pi_i(\cdot|s')$.

$$Q_{\text{tar}}(s, a) = c + \gamma \left( \min_{d=1,2} Q_{\bar{\theta}_d}(s', a' \sim \pi_i(\cdot|s')) - \alpha \log \mu_\phi(\bar{a}|s') \right), \quad \bar{a} \sim \mu_\phi(\cdot|s'),$$

where $\alpha$ is the learned entropy coefficient and $\mu_\phi$ is the parametrized behavior policy. The variance target update, $M_{\text{tar}}$, then incorporates the Q-value temporal difference (TD) error, $\delta_Q = Q_{\text{tar}} - \min_{d=1,2} Q_{\theta_d}(s, a)$,

$$M_{\text{tar}}(s, a) = \delta_Q^2 + \gamma^2 \left( \min_{d=1,2} M_{\bar{w}_d}(s', a' \sim \pi_i(\cdot|s')) - \alpha \log \mu_\phi(\bar{a}|s') \right).$$

To ensure the behavior policy does not diverge from target GVF policies $\pi_i$, we introduce a KL regularization term that maintains alignment with each target policy by minimizing $\text{KL}(\mu_\phi(\cdot|s) \| \pi_i(\cdot|s))$.

As computing exact Q-values is infeasible for continuous state-action domains, we approximate these values using Monte Carlo (MC) rollouts over 100 episodes for a fixed set of randomly selected 50 sampled state-actions. This MC approximation enables accurate computation of MSE by comparing learned Q-values with empirical MC estimates.

Finally, the behavior policy $\mu_\phi$ is updated to minimize the MSE by focusing on areas of high variance, effectively improving estimation efficiency. Further, the entropy term is similar to standard SAC algorithm for improving the exploration. This update step, driven by the variance network, is computed as:

$$\nabla_\phi \sum_{s \sim \mathcal{D}} \left( \min_{d=1,2} M_{w_d}(s, \bar{a}) - \alpha \log \mu_\phi(\bar{a}|s) + \beta \sum_{i=1}^N \text{KL}(\mu_\phi(\cdot|s) \| \pi_i(\cdot|s)) \right),$$

where action $\bar{a}$ is sampled from $\mu_\phi(\cdot|s)$. To support adaptive exploration, we update $\alpha$ by optimizing:

$$-\alpha \log \mu_\phi(\cdot|s) + \bar{H},$$

where $\bar{H}$ is the target entropy.

With these adaptations, SAC-`GVFExplorer` effectively estimates GVFs values in continuous state-action domains, achieving lower MSE compared to baselines such as `RoundRobin` and `UniformPolicy` and performing well in complex Mujoco environments.

We used TD3 to train target policies for each GVF and selected mid-level performing policies as target policies for the respective GVF tasks. This setup allows the behavior policy to efficiently gather data for parallel GVF estimation.

**Algorithm 2:** SAC-based `GVFExplorer`

---

**Input:** Target policies $\pi_{i \in \{1,\ldots,n\}}$, initialized behavior policy $\mu_\phi$, replay buffer $\mathcal{D}$, primary networks $Q$ with $\theta_1, \theta_2$, primary $M$ variance with $w_1, w_2$, target networks $Q_{\bar{\theta}_1}, Q_{\bar{\theta}_2}$, $M_{\bar{w}_1}, M_{\bar{w}_2}$, learning rates $\alpha_Q, \alpha_M$, mini-batch size $b$, entropy coefficient $\alpha$, KL regularizer $\beta$, update frequencies $p$, $m$, $l$, target entropy $\bar{H}$, training steps $K$

1   **for** *environment step* $k = 1, \ldots, K$ **do**
2      Select action $a \sim \mu_\phi(\cdot|s)$
3      Observe next state $s'$ and cumulants $c$
4      Store transition $(s, a, s', c)$ in replay buffer $\mathcal{D}$
5      **if** $step\%p == 0$ **then**
6          Sample mini-batch $\mathcal{D} \sim (s, a, s', c)$
7          //Q-critic update
8          Compute $Q_{tar}(s,a) = c + \gamma \left( \min_{d=1,2} Q_{\bar{\theta}_d}(s', a' \sim \pi_i(\cdot|s')) - \alpha \log \mu_\phi(\bar{a}|s') \right)$, $\bar{a} \sim \mu_\phi(\cdot|s')$
9          Update $Q_\theta$ with MSE loss: $(Q_{tar} - Q_{\theta_d}(s,a))^2$ for $d = 1, 2$
10         //Compute TD error
11         $\delta_Q = Q_{tar} - \min_{d=1,2} Q_{\theta_d}(s,a)$
12         //Variance-critic update
13         Compute $M_{tar}(s,a) = \delta_Q^2 + \gamma^2 \left( \min_{d=1,2} M_{\bar{w}_d}(s', a' \sim \pi_i(\cdot|s')) - \alpha \log \mu_\phi(\bar{a}|s') \right)$, $\bar{a} \sim \mu_\phi(\cdot|s')$
14         Update $M_w$ with MSE loss: $(M_{tar} - M_{w_d}(s_t, a_t))^2$ for $d = 1, 2$
15      **end**
16      **if** $step\%l == 0$ **then**
17          Update target networks: $\bar{\theta}_d = \theta_d$, $\bar{w}_d = w_d$
18      **end**
19      **if** $step\%m == 0$ **then**
20          // Update behavior policy $\mu_\phi$
21          Update $\phi$ using
            $\nabla_\phi \sum_{s \sim \mathcal{D}} \left( \min_{d=1,2} M_{w_d}(s, \bar{a}) - \alpha \log \mu_\phi(\bar{a}|s) - \beta \sum_{i=1}^{N} \mathrm{KL}(\mu_\phi(\cdot|s) \parallel \pi_i(\cdot|s)) \right)$,
22          where $\bar{a}$ is sampled from $\mu_\phi(\cdot|s)$.
23          //Update $\alpha$ entropy regularizer
24          Update $\alpha$ with loss $(-\alpha \log \mu_\phi(\cdot|s) + \bar{H})$
25      **end**
26   **end**
27 **Returns** Estimated GVF values $Q_\theta^i(s, \cdot)$ for $i = \{1, \ldots, n\}$

---

