# OpenReview forum: "Adaptive Exploration for Data-Efficient General Value Function Evaluations"
_NeurIPS.cc/2024/Conference — NeurIPS 2024 poster_

### Official Review · Reviewer_3wne · 2024-07-10

**Soundness:** 3
**Presentation:** 3
**Contribution:** 2
**Rating:** 6
**Confidence:** 4

**Summary:**

This paper presents a novel method named GVFExplorer for efficiently evaluating multiple Value Functions with different policy (GVFs) in parallel using off-policy methods. It adaptively learns a single behavior policy that minimizes the total variance in return across GVFs, thus reducing required environmental interactions. The method uses a temporal-difference-style variance estimator and proves that each behavior policy update decreases the overall mean squared error in GVF predictions. The performance of GVFExplorer is empirically validated in various settings, including tabular and non-linear function approximation, with stationary and non-stationary reward signals.

**Strengths:**

- idea of choosing behavior policy for simultaeneous learning of multiple policies is quite novel, and seem to be able to be applied to some applications.
- systematic way of deriving algorithm is novel and interesting, and makes sense as well

**Weaknesses:**

- More experiments would help understand the behavior of the algorithm.
  - uniform policy being the best baseline does not seem to be a good baseline choice. Having some more competetive baselines, e.g. ablations, would have been much better.
  - experimented environments seem too synthetic. would like to see results on typical (and challenging) RL environments, e.g., mujoco.

**Questions:**

- The paper starts with minimizing MSE, and argues that it uses unbiased IS estimation, leading to a policy that minimizes the variance of return. However, GVF in algorithm is estimated with $Q_\theta$, which has been learned with expected Sarsa, which is a baised estimate given that the target contains $Q_\theta$ instead of GT $Q^\pi$. How do we ensure that the analysis above also works in algorithm 1? What happens if our function approximator is crude so that the bias is large?

- Why does PER drastically improve the performance of GVFExplorer? If we have infinite number of data in our experience replay, GVFExplorer would have very similar effect as PER. In my opinion, it seems more natural to have reduced effect of PER as we are sampling hard states more with GVFExplorer, and it is what PER tries to do as well. (it also does not go well with the rest of the paper, as PER is about training efficiency where GVFExplorer is about the choice of behavior policy)

**Limitations:**

The authors adequately addressed the limitations.

---

> ### Author Rebuttal · Authors · 2024-08-07
>
> Thank you for your detailed feedback and valuable experimental suggestions, which we have now incorporated into our rebuttal. Based on your input, we have also **added results in Mujoco in the main comment (refer Fig 1 in PDF)**. We appreciate your recognition of the novelty of adaptively learning the behavior policy to efficiently evaluate multiple GVFs in parallel and acknowledgment of the systematic derivation of the algorithm. Detailed responses to your queries are provided below.
>
> 1. **"How can Uniform sampling be a competitive baseline?"**
>
>    We use several baselines: uniform, mixture policy, round-robin policy, SR-based policy, and BPS. Learning a behavior policy for multiple GVFs is relatively unexplored, making it challenging to identify more baselines. Our empirical results indicate that the relative performance of these baselines can vary depending on target policy characteristics.
>
>    For example, in Fig. 7 of the main paper, round-robin and mixture policies perform better when using semi-greedy fixed target policies. Conversely, the uniform policy can be advantageous when the target policies exhibit low goal-reaching probability, because it helps hit the goal by chance.
>
> 2. **"Query: Show experiments in Mujoco"**
>
>    We have now expanded our algorithm to continuous actions and shown results in Mujoco. Please refer to **Fig 1 in the attached PDF** and the main comment for the graphs and explanation of the Mujoco experiments. We believe these additions significantly strengthen our paper and provide a more comprehensive evaluation of our proposed method.
>
> 3. **"How does using an approximated Q in the target, rather than the true Q, affect our algorithm's analysis? What is the impact on MSE performance when the function approximator is crude?"**
>
>    All TD-based methods, including IS-TD, are biased because of bootstrapping. Only with MC return, IS is unbiased. We use the unbiased estimator as a motivation to reduce the problem to minimizing the variance. Empirically, we used TD-based methods in all experiments to approximate both value and variance, due to the better stability and efficiency of these estimators in large-scale problems.
>
>    To better understand the impact of using a crude function approximator on MSE, we have now added experiments on this ablation study. Specifically, we examined the average MSE in a 20x20 grid with two distinct GVFs. We produced features by reducing the state space into a 10x20 feature grid (grouping factor = 2) and a 5x20 feature grid (grouping factor = 4). An approximation factor of 1 shows the result without any function approximation.
>
>    In **Fig 2 (PDF attached in main comments)**, as expected, the overall MSE increases with cruder approximations. Despite this, GVFExplorer outperforms round-robin and mixture policies. However, when the approximation is very crude (factor = 4), the uniform policy performs better due to poor variance estimates. These results suggest that GVFExplorer is robust with reasonable function approximators, but can degrade with extremely coarse ones, which is to be expected.
>
>    Further, these results are also strengthened by the performance in the attached Mujoco environment (Fig 1 in PDF in main comments) where function approximators are used.
>
> 4. **"How does PER improve GVFExplorer performance? How is PER different from our algorithm? Given infinite data in the replay buffer, would GVFExplorer have the same effect as PER?"**
>
>    PER complements GVFExplorer by enhancing data efficiency. While GVFExplorer optimizes the behavior policy to sample informative data, PER reweights the collected samples in the buffer according to priority, ensuring that the mini-batches sampled for gradient descent are selected efficiently. Please refer to lines 307-309 for reasoning on performance boosts of GVFExplorer with PER.
>
>    With infinite data, the impact of both GVFExplorer and PER would diminish, and direct sampling from the target policies would suffice. However, in practical settings with limited data, GVFExplorer's ability to actively influence data collection provides a significant advantage.
>
>    **It is crucial to distinguish between GVFExplorer and PER**. GVFExplorer adapts the behavior policy based on variance estimates, while PER reweighs the sampling of the existing data already collected within the buffer. We used PER with all algorithms, including baselines, as shown in Fig 3a of the main paper; all baselines, except the SR method, show improved performance with PER.

---

> > ### Comment · Reviewer_3wne · 2024-08-12
> >
> > Thanks for the response. I raised my score.

---

> > > ### Author Response · Authors · 2024-08-13
> > > **Author response to Reviewer 3wne**
> > >
> > > Hello Reviewer, thanks for responding and raising the score.

---

### Official Review · Reviewer_A7c7 · 2024-07-11

**Soundness:** 4
**Presentation:** 3
**Contribution:** 3
**Rating:** 7
**Confidence:** 4

**Summary:**

This paper propose a new algorithm to solve the general value function evaluations problem. In essence, GVFs can be seen as high dimensional value functions. The authors propose a temporal difference learning algorithm that minimizes the overall variance in the return distribution, in the hope to improve the behavior policy for better exploration, such that the samples the behavior policy produces better suffices for off-policy evaluations for each GVFs.

**Strengths:**

1. The idea of minimizing the variance of the return of the behavior policy is novel. While the idea might now be groundbreaking, it is new in this field.
2. Incorporating a temporal difference to approximate the overall variance of the return of the each general values are interesting. Indeed, for large scale problems, TD is a better solution overall.
3. The derivation and analysis of the algorithm is sounds and rigorous.
4. Overall the paper is clearly presented.

**Weaknesses:**

1. The core idea behind this paper is to propose a solution to solve the data collection problem. This paper does answer the question of how to minimize the variance of the returns, but fail to convince me entirely why we should do that at the first place, either through proofs, or empirical investigation. I think this is the biggest weakness of the paper.
2.  The experiment section is nice and clear but the problem class is a bit simple (gridworld). For a paper without strong theoretical results, experiments are usually expected to have more materials. In this sense, the results are not very convincing.

**Questions:**

Please see weakness above.

**Limitations:**

Yes.

---

> ### Author Rebuttal · Authors · 2024-08-07
>
> Thank you for the detailed review and positive feedback on our algorithm's rigorous derivation and analysis. Based on your suggestion, we have now **added Mujoco experiments result in the main comment**. We respond to each query below.
>
> 1. **“Why did we choose to minimize variance of return as our objective”?**
>
>    We think there may be a misunderstanding regarding the question. Below, we have tried to clarify our rationale, but please let us know if further discussion would be beneficial.
>
>    The primary objective is to identify a behavior policy that minimizes MSE when evaluating multiple GVFs. As outlined in lines 120-125, $\text{MSE} = \text{bias}^2 + \text{variance}$. Using the unbiased IS estimator, we reduce the problem to minimizing the variance, which is the core objective. Minimizing variance improves the accuracy of value estimates by reducing uncertainty. This ensures that the behavior policy collects the most informative samples, thereby reducing estimation error quickly.
>
>
>    Furthermore, Owen et al. (2013) demonstrated that using a minimum-variance optimal behavior policy ($\mu^*$) for a single target policy ($\pi$) in scenarios with known dynamics can lead to performance improvements. The value obtained under $\mu^*$ is greater than under $\pi$, where the improvement is directly related to the variance reduction achieved. This indicates that the higher the variance reduction under $\mu^*$ policy, the more significant the performance improvement. We hypothesize that similar effects could be observed in multiple GVF policy “Control” scenarios, which we aim to investigate in future research. This work lays the foundations under the policy evaluation context.
>
>    We will include this rationale in the camera-ready version to further substantiate our approach.
>
> 2. **“Query regarding more experiments in Mujoco”**
>
>    We have now added the empirical results in the Mujoco environment with continuous actions. Please refer to the attached PDF and the main comment for the performance graphs and explanation on Mujoco experiments.

---

> > ### Comment · Reviewer_A7c7 · 2024-08-11
> > **Thank you for the response**
> >
> > I thank the authors for the response. Judging from other reviews and the added experiments, I will keep the same score for now.

---

### Official Review · Reviewer_pomH · 2024-08-08

**Soundness:** 4
**Presentation:** 4
**Contribution:** 3
**Rating:** 6
**Confidence:** 4

**Summary:**

This paper presents a new algorithm for collecting data needed to learn multiple GVFs in parallel. By focusing data collection on high-variance (s,a) pairs, an agent is able to collect data that will reduce the variance of estimated GVFs. The authors contribute a sort of contraction-mapping proof that using their algorithm will result in non-increasing variances.

**Strengths:**

The algorithm is simple, with some reasonable theoretical properties.
The paper addresses a good problem.
The paper is very well written
The experiments seem reasonable and informative
The method performs well against other baselines

**Weaknesses:**

I have one minor-but-important quibble about the paper. In Thm. 4.2, the authors prove that aggregated variance is "<=" upon successive iterations. However, in the english description of the result, the authors state that the aggregated variances "decrease with each update step".

This is NOT what you proved - you proved that aggregated variance "does not increase."  The same claim is made in the abstract, and again in the conclusion.  I think it's important to be clear on this point, so I would ask the authors to rephrase this.

**Questions:**

I wonder if there are potential degeneracies in the algorithm. For example, if a behavior policy never tries a certain action, it seems like the M's for that action will be 0, and the new behavior policy will assign 0 probability to taking that action in the future, leading to a situation where you never get the data you need.  Similarly, if a certain state is never visited (in the tabular case), would the estimate of M be zero?

If so, does that imply that there is an unstated constraint on the initial behavior policy -- something about exploring states and actions sufficiently?

A related question: if the cumulant function is sparse, is it possible to not get enough non-zero data to get non-zero M estimates?

(by the way, it's these sorts of questions where I can kind of see that degeneracies may arise that will never get ironed by your algorithm, which is why the difference between "<" and "<=" in your proof is important)

**Limitations:**

It seems like a central limitation of the work is a disconnect between what was theoretically proven (which relies on a perfect knowledge of the variances M), and what will happen in practice (the M's must be estimated).

It would be nice if the paper outlined what happens to the algorithm as a function of the error in the estimates of M.

---

> ### Author Response · Authors · 2024-08-09
> **Author's Response to Reviewer pomH**
>
> Thank you reviewer for your thoughtful and constructive feedback. We provide a detailed response to the asked questions below.
>
> 1. **“Clarification regarding Theorem 4.2”**
>
> We appreciate your careful attention, you are correct that Theorem 4.2 demonstrates that the aggregated variance "does not increase" with each update, rather than strictly "decreases." due to "<=" sign in the proof. We will make the necessary revisions to the paper to accurately reflect this.
>
> 2. **“Regarding question on potential degeneracies and exploration of algorithm”**
>
> Thank you for raising this important point. Just like in standard RL, to obtain reliable estimates—whether for Q-values or variances—it's important to have at least initial exploration of different areas of the state-action space. This can be achieved by incorporating any standard initial exploration technique.
>
> In our work, the proposed algorithm assumes sufficient initial exploration to gather necessary data. We address this by initializing M values to a non-zero constant across all state-action pairs and also allowing epsilon exploration, where epsilon decays over time. This ensures agents visit a wide range of state-action pairs early on, preventing issues of zero variance for unvisited state-action pairs. We added epsilon-exploration for all algorithms including baselines.
>
> 3. **“Effects on M with sparse cumulant”**
>
> In scenarios with sparse cumulants, the non-zero initialization of M ensures that all states are visited, providing a fair opportunity to correct M estimates over time.
>
> 4.  **“What happens to the algorithm as a function of the error in the estimates of M?”**
>
> Similar to any TD-based algorithm, the empirical version of our approach relies on initial estimates, which will be imperfect. If the initial M estimates are incorrect, the TD error will indicate this, either positively or negatively. The behavior policy update will then be influenced by these M estimates, leading the agent to gather new samples as it interacts with the environment. As agent collects more data, the M estimates will improve—either increasing or decreasing for specific states—allowing the behavior policy to adjust and correct its sampling strategy accordingly. If an M estimate is very small compared to its true value, the agent will first focus on states with higher M estimates, correct those values, and then revisit the poorly estimated states to update their M estimates. This iterative process is analogous to TD-based Q-learning updates in standard RL.
>
> Additionally, if the target policies have non-zero probabilities, our behavior policy incorporates a small epsilon probability over those state-action pairs. This approach is also supported by Lemma 6.1 which requires a bounded difference between the behavior and target policies to ensure that the variance function remains well defined. For Mujoco experiments, we added KL term to limit this divergence.
>
> We will include these explanations in the camera-ready version to provide a practical understanding of our algorithm.

---

### Author Rebuttal · Authors · 2024-08-07

Thank you for the valuable and constructive feedback. We are encouraged by your recognition of the *novelty of the problem* and acknowledgment of our *algorithm derivations as systematic and rigorous*.

Based on your feedback, we have extended our experimental results in the **Mujoco environments (Fig 1)** and included these results here. We have also added the codebase to the code repository. Further, based on Reviewer 2's suggestion, we are including an ablation study on the effect of using **feature approximator on the MSE performance metric (Fig 2)**. Both the results in Mujoco and the ablation study further support the benefits of using GVFExplorer for data-efficiently evaluating multiple GVFs in parallel.

$~~~~~~~~~~~$
## Mujoco Experiments (Reviewer 1 and 2)

We have now conducted additional experiments using the DM-control suite to experiment with the Mujoco “walker” and “cheetah” domains. For the walker, we define two distinct GVFs: walk and flip. For the cheetah, we also evaluate two GVFs: walk and run. To extend the implementation in a continuous action environment, any policy gradient algorithm can be used. Similar to the Q-value critic, we implement a separate M-variance critic using neural networks for both. The behavior policy interacts with the environment and gathers samples. It uses the samples to first update the two critics, followed by an update to the behavior policy network to minimize the MSE (our objective). We also added a KL regularizer between the behavior policy and the two target policies of the GVFs to prevent divergence. We use Monte Carlo estimates of the true GVF values and compare the MSE between these values and the output of the Q-critic network. We use the same Q-critic architecture for all algorithms, including the baselines.

**Fig 1 of the attached PDF:** GVFExplorer significantly reduces the average MSE compared to baselines such as RoundRobin and UniformPolicy, demonstrating that an adaptive behavior policy collects more informative samples.

$~~~~~~~~~~~$
## Ablation study on effects of using feature approximator on MSE (Reviewer 2)

We conducted an ablation study to understand the effects of using a very coarse function approximator on MSE. We evaluated the averaged MSE over two distinct GVFs in a 20x20 grid. To simulate approximation, we mapped each 2x1 grid region to the same feature, resulting in a grouping factor of 2 and a 10x20 feature grid. Similarly, mapping each 4x1 grid region to the same feature resulted in a grouping factor of 4 and a 5x20 feature grid. As shown in **Fig 2 in attached PDF**, the overall MSE increases with cruder approximations, as expected. GVFExplorer generally outperforms the baselines, but with a very crude approximation (factor = 4), the uniform policy performs better due to poor variance estimates. These results highlight GVFExplorer's robustness with reasonable function approximators. More details of the experiment are presented in R2’s response.

---

### Decision · Program_Chairs · 2024-09-25

**Decision:**

Accept (poster)

**Comment:**

The reviewers and I generally liked this paper. While the core algorithm is a bit obvious, the paper is well-written and makes a positive contribution to the RL literature on GVFs.  The paper has a nice mix of theory and empirical work, and addresses a problem that has not been previously studied, and provides proofs of correctness (never easy to do in an RL setting!).  The paper also provides a clear, well-written discussion of the issues surrounding GVFs.  Based on the alignment of all of these positive factors, I recommend acceptance.